



# Use of an observation-based aerosol profile in simulations of a mid-latitude squall line during MC3E: Similarity of stratiform ice microphysics to tropical conditions

Ann M. Fridlind[1], Xiaowen Li[2,3], Di Wu[3,4], Marcus van Lier-Walqui[1,5], Andrew S. Ackerman[1], Wei-Kuo Tao[3], Greg M. McFarquhar[6], Wei Wu[6], Xiquan Dong[7], Jingyu Wang[7], Alexander Ryzhkov[8], Pengfei Zhang[8], Michael R. Poellot[9], Andrea Neumann[9], and Jason M. Tomlinson[10]

[1]NASA Goddard Institute for Space Studies, 2880 Broadway, New York, NY, USA
[2]Morgan State University, Baltimore, MD, USA
[3]NASA Goddard Space Flight Center, Greenbelt, MD, USA
[4]Science Systems and Applications, Inc., Lanham, MD, USA
[5]Columbia University, New York, NY, USA
[6]University of Illinois, Urbana-Champaign, IL, USA
[7]University of Arizona, Tucson, AZ, USA
[8]Cooperative Institute for Mesoscale Meteorological Studies, University of Oklahoma, and National Severe Storms Laboratory, Norman, OK, USA
[9]University of North Dakota, Grand Forks, ND, USA
[10]Pacific Northwest National Laboratory, Richland, WA, USA

*Correspondence to:* Ann Fridlind (ann.fridlind@nasa.gov)

**Abstract.**

Advancing understanding of deep convection microphysics via mesoscale modeling studies of well-observed case studies requires observation-based aerosol inputs. Here we derive hygroscopic aerosol size distribution input profiles from ground-based and airborne measurements for six convection case studies observed during the Midlatitude Continental Convective

5    Cloud Experiment (MC3E) over Oklahoma. We demonstrate use of the aerosol inputs in mesoscale model simulations of the only well-observed case study that produced extensive stratiform outflow, on 20 May 2011. At well-sampled elevations between $-10$ and $-23°C$ over widespread stratiform rain, ice crystal number concentrations are consistently dominated by a single mode near $\sim 400\ \mu$m in randomly oriented maximum dimension ($D_{max}$). The ice mass at $-23°C$ is primarily in a closely collocated mode, whereas a mass mode near $D_{max} \sim 1000\ \mu$m becomes dominant with decreasing elevation to the $-10°C$ level,

10    consistent with possible aggregation during sedimentation. However, simulations with and without observation-based aerosol inputs systematically overpredict mass peak $D_{max}$ by a factor of 3–5 and underpredict ice number concentration by a factor of 4–10. Previously reported simulations with both two-moment and size-resolved microphysics have shown biases of a similar nature. The observed ice properties are notably similar to those reported from recent tropical measurements. Based on several lines of evidence, we speculate that the microphysics pathways associated with deep tropical convection outflow also occurred

15    in the 20 May MC3E case, likely associated with warm-temperature ice multiplication that is not well understood or well represented in models.



## 1 Introduction

The impacts of hygroscopic, absorbing, and ice-nucleating aerosol on deep convection have been the subject of intensive study using both observational and modeling approaches, as summarized in several recent reviews (e.g., Tao et al., 2012; Wang, 2013; Altaratz et al., 2014). A hindrance for the modeling studies is the widely reported finding that different advanced microphysics

schemes, given the same environmental conditions and setup, often predict substantially differing results in terms of ice mass mixing ratios and other cloud properties (e.g., Zhu et al., 2012; Van Weverberg et al., 2013; Fan et al., 2015; Wang et al., 2015b; Tao et al., 2016). Microphysics schemes give such diverse results at least in part owing to the complexity of updraft microphysics and a paucity of existing field and laboratory data adequate to constrain all of the relevant physical processes and parameters (e.g., Zeng et al., 2011; Varble et al., 2014a).

One important objective for the representation of convective updraft microphysics in climate models, and by extension in the higher-resolution simulations commonly used to help develop climate model parameterizations (e.g., Wu et al., 2009; Storer et al., 2015; Wong et al., 2015), is confidence in the simulation of convective updraft outflow ice properties, which may substantially impact global radiative budgets and circulation (e.g., Houze, 2004; Schumacher et al., 2004; DelGenio, 2012; Mauritsen and Stevens, 2015). Based on comparison of large-domain convection-permitting simulations with tropical satellite

data, Van Weverberg et al. (2013) concluded that such simulations are sensitive to microphysics parameterizations and that more complex schemes do not necessarily perform better. Evidence from recent tropical field measurements has indicated that microphysics schemes could be failing to represent efficient ice multiplication that may strongly impact tropical updraft glaciation rate, outflow ice size, and precipitation efficiency (Ackerman et al., 2015), providing further motivation to advance fundamental knowledge of updraft microphysical pathways. Owing to the challenging complexity of coupled dynamical and

microphysical processes within outflow-generating updrafts and the increasing ability of computational approaches to resolve such coupling (e.g., Lebo and Morrison, 2015), the goal of improving understanding of deep convection processes through high-resolution simulation of well-observed case studies is increasingly attractive (e.g., Yang et al., 2015).

Establishing reliability of high-resolution simulations to advance fundamental knowledge of convective microphysics depends on observational constraint of initial conditions as well as simulation results. Whereas thermodynamic conditions may

be well characterized by routine observations or reanalysis fields (e.g., Zhu et al., 2012), aerosol initial conditions for any observed case study generally require detailed observational inputs (e.g., Yang et al., 2015). Here we develop hygroscopic aerosol input data sets for six convection events that were well observed during the Midlatitude Continental Convective Cloud Experiment (MC3E), a joint field program of the U.S. Department of Energy Atmospheric Radiation Measurement (ARM) program and the NASA Global Precipitation Measurement Mission (Jensen et al., 2016). Aerosol input profiles are archived as

Supplement 1. We also demonstrate use of derived aerosol input size distributions in simulations of the only event with extensive stratiform outflow that was well-sampled by aircraft, on 20 May 2011 (Wang et al., 2015a; Wu and McFarquhar, 2016), with an emphasis on comparing simulated hydrometeor size distributions with observations. Enabling accurate simulation of such long-lived and radiatively important stratiform outflow is a valuable goal for global models (e.g., Lin et al., 2012).



In the following sections, we describe selection of six convection case studies from the MC3E campaign (Section 2), derivation of aerosol specifications for each case from ground-based and aircraft measurements (Section 3), and comparison of simulated hydrometeor size distributions with observations for the 20 May case study (Section 4). Results are summarized and discussed in the context of other recent measurement campaigns and modeling studies (Section 5).

## 2 Case studies

The MC3E domain (Fig. 1) is defined by a sounding array containing a triangular X-band radar array and a central facility with additional instruments, including a Ka-band ARM Zenith Radar (KAZR), a NOAA S-band (2.8-GHz) profiling radar, a TSI model 3010 Condensation Particle Counter (CPC), a DMT model 1 Cloud Condensation Nuclei (CCN) counter, and a Tandem Differential Mobility Analyzer (TDMA). We begin by focusing on the 22 April–25 May 2011 time period of MC3E for which a large-scale forcing data set was initially derived using a variational analysis approach (Jensen et al., 2015). During this time period, ten flights of the University of North Dakota Citation aircraft provide profiles of aerosol properties to elevations of 8 km or higher (on 22, 25, and 27 April, and 1, 10, 11, 18, 20, 23, and 24 May). Aerosol number size distribution in the 0.06–1-$\mu$m diameter size range was measured on the Citation with an Ultra High Sensitivity Aerosol Spectrometer (UHSAS) and the number concentration of aerosols with diameter larger than 10 nm was measured with a TSI 3771 Condensation Particle Counter (CPC).

Owing to the importance of identifying fine-scale convection structural features in simulations, we first select case studies when the C-band Scanning ARM Precipitation Radar (C-SAPR) was fully or partly operational, which eliminates two flight dates (10 and 11 May). In order to allow simultaneous use of profiling instruments, we focus on cases in which substantial convection features passed directly over the KAZR and other nearby instruments at the central facility, which eliminates two more flight dates (22 April and 18 May). This leaves six flight dates for which aerosol property profiles are derived here for use in convection simulations: 25 and 27 April and 1, 20, 23, and 24 May. Figure 2 illustrates the varying convection that passed over the central facility on each date, including the long duration of stratiform rain following deep convection in the 20 May case.

## 3 Aerosol input data

Based on evidence that aerosol consumption may be efficient in strong updrafts (e.g., Fridlind et al., 2004; Yang et al., 2015) and nanometer-sized particles could be be nucleated (e.g., Ekman et al., 2006; Khain et al., 2012), emphasis is placed on deriving size distribution profiles that include aerosol of all available sizes for each case. Owing to lack of measurements, we unfortunately omit coarse-mode (supermicron) aerosol, which may constitute ∼1–10 cm$^{-3}$ aerosol that are a small fraction of relevant hygroscopic aerosol but may be especially relevant to heterogeneous ice nucleation (DeMott et al., 2010; Corr et al., 2016). To make simulation specifications relatively simple, it is also assumed that a single size distribution profile will be used





in each case (no time dependence of specified environmental aerosol conditions), as in past deep convection studies that have specified observation-based aerosol profiles (e.g., Barth et al., 2007; Fridlind et al., 2012; Yang et al., 2015).

As shown in Fig. 3, the ground-based aerosol instrumentation operated continuously with few interruptions throughout the campaign. For each case study a two-hour time period prior to the detection of surface precipitation at the central facility is first

identified on each date (dotted vertical lines in Fig. 3). When averaging measurements over these pre-convection periods, we find that the total aerosol number concentration reported by the DMA (0.012–0.74 $\mu$m dry diameter) agrees with that reported by the CPC (0.01–3 $\mu$m) to within 30% in all cases except 25 April and 1 May, when the DMA concentration is 80% higher and 50% lower, respectively. The reasons for disagreement are unclear; here we will rely on the DMA data for size distribution information, while noting the discrepancy.

Also shown in Fig. 3 is aerosol hygroscopicity parameter ($\kappa$) derived from HTDMA measurements, linearly averaged in six reported size ranges. Commonly low $\kappa$ values of $\sim$0.1 are consistent with those derived from airborne aerosol size distribution and CCN measurements at a similar time of year over the Southern Great Plains site (Vogelmann et al., 2015). Similar to long-term measurements from the organics-rich Amazon rain forest (Pöhlker et al., 2016), there appears to be a common trend of increasing $\kappa$ with size between the Aitken and accumulation mode size ranges.

In general, the variability of spread between CCN, HTDMA, and CPC indicate that nucleation mode aerosols smaller than 0.1 $\mu$m in dry diameter commonly appeared in large concentrations, but were also commonly absent. CCN data reported at the highest supersaturation measured (slightly above 1%) variably account for roughly 15–80% of the aerosol reported by the CPC, and range over nearly an order of magnitude ($\sim$400–3000 cm$^{-3}$) across the six case studies, with an intermediate value of $\sim$2000 cm$^{-3}$ on 20 May.

The MC3E aircraft measurements were commonly taken during precipitation at the ground site in order to sample cloud and precipitation conditions (cf. Fig. 3). We filter all aircraft aerosol measurements to remove in-cloud samples by imposing the stringent requirement that hydrometeors in the 2–50-$\mu$m diameter range measured by a Cloud Droplet Probe (CDP) remain below the detection limit (cf. McFarquhar and Cober, 2004), which is roughly 0.03 cm$^{-3}$ given the CDP sample area of 0.3 mm$^2$ (Lance et al., 2010) and a typical Citation aircraft speed of 100 m s$^{-1}$. Unfortunately, out of the six convection

case studies considered here, UHSAS data were available only for the first three and CPC data only for the latter three. After surveying the available CPC and UHSAS data for the six case studies, we therefore analyze aerosol measurements from all flight dates to provide estimates of missing information.

Out-of-cloud CPC profiles measured on 20, 23 and 24 May indicate that nucleation mode aerosols could be present in the region even when they were not detected at the ground site during the pre-convection period (Fig. 4). Based on the CPC data

available from twelve flights during MC3E, freshly nucleated particles were commonly associated with condensation nuclei concentrations in excess of $10^4$ cm$^{-3}$, typically limited to or most concentrated below 1–3 km in altitude, and encountered during every flight except that on 23 May. Thus, even when not present at the ground site, as on 20 May, nucleation mode particles were virtually always present somewhere nearby. However, aircraft data consistently indicated a high degree of variability in the distribution of nucleation mode particles in the boundary layer. Maps of CPC concentration as a function of latitude

and longitude on each flight indicated that the nucleation mode was commonly limited to a short flight segment (not shown),



indicative of transects through plumes likely generated by emissions from multiple nearby power plants that may not broaden efficiently downwind (e.g., Wang et al., 2006; Stevens et al., 2012). The airport could also be a source affecting the airborne samples (e.g., Westerdahl et al., 2008).

Out-of-cloud UHSAS profiles measured on 25 and 27 April and 1 May indicate median concentrations of 100–1000 cm$^{-3}$

commonly decreasing with increasing elevation (Fig. 5). On the latter two dates, long flight legs at a single elevation indicate horizontal variability commonly exceeding an order of magnitude both greater and lesser than relatively well-defined mean profiles.

Fig. 6 shows the profiles of median UHSAS and CPC concentrations on three non-case-study dates for which both instruments functioned. UHSAS is shown to represent a fraction of CPC that generally decreases with height above ∼2 km,

consistent with the expectation that the surface is a source of the larger aerosol. Below 2 km, however, UHSAS/CPC again sometimes decrease, not because UHSAS decreases but because CPC increases, consistent with evidence that the surface is also a source of fine particles. Considering the general vertical trend of number concentration during the seven MC3E flights for which both UHSAS and CPC data are simultaneously available, median out-of-cloud UHSAS number concentration summed over 0.06–1.0 $\mu$m dry diameter accounts for 20–60% of collocated median CPC number concentration when taken over 1-km

vertical layers during each flight (Fig. 7). The campaign-wide median profile of the ratio of UHSAS to CPC is archived as Supplement 2.

Since each case study date offered only one of the two instruments, the median ratio of UHSAS to CPC number concentration shown in Fig. 7 is used as a guide for scaling ground-based measurements, which are derived as follows. First, all DMA size distributions measured during each pre-convection period are fit with lognormal modes using the approach described by

Vogelmann et al. (2015). It is found that two to three modes provide the best fit for each size distribution (Fig. 8), where the nucleation-mode fits are intended for truncation at a user-determined size guided by model representation of ambient molecular clusters (cf. Jiang et al., 2011). The mode properties are then averaged in time. In the case that three modes provided a best fit, those are each averaged by mode. In the case that only two modes provide a best fit, then if the mode geometric mean diameter of the smaller mode is smaller than or equal to 0.01 $\mu$m, that mode is considered the smallest of three; otherwise,

that mode is considered the middle of three. A simple linear average of the modal properties (number concentration, geometric mean diameter and standard deviation) is then adopted in each of the three modes. A hygroscopicity parameter ($\kappa$) is then derived for each mode as follows. First a $\kappa$ value is calculated from the mean growth factor measured by the TDMA during each pre-convective period, available in six size cuts over 0.013–0.40 $\mu$m in dry diameter. Then a $\kappa$ value is assigned to each DMA size bin, using linear interpolation. Finally, a $\kappa$ value is calculated for each mode as a number-weighted average over

occupied bins (cf. Fig. 8).

The ambient aerosol modal properties are treated as constant in space and time for each case study. However, the number concentration in each mode is assumed to vary with height, and its variation is derived from aircraft measurements as follows. First, we adopt the ground-based three-mode fit for each case study as representative of the bottom kilometer of the atmosphere. Owing to an absence of fine-mode aerosol size distribution information aloft, we then assume that aerosol in the smallest mode





are confined to the surface layer, consistent with the occurrence of increased concentrations primarily near the surface (Fig. 4). Numbers in the larger two modes above the surface layer are then determined as follows.

If only UHSAS data are available, for each 1-km layer above the surface layer the number concentration in the larger mode is set to the total concentration measured by UHSAS, and the number concentration in the smaller mode is set such that the ratio of UHSAS to total assumed aerosol matches the experiment-wide median ratio at that altitude. Number concentrations in any of the smallest few UHSAS bins that exceeded five times the concentration in bins with diameter larger than 0.1 $\mu$m (in terms of dN/dlogD) appeared spurious, and these are neglected when present (e.g., lowest two levels in Fig. 9). The resulting number-wise scaling of the ground-based DMA size distributions to the total UHSAS concentration often gives remarkably close fits to bin-wise median UHSAS size distributions above 1 km, although aircraft data appear to be variably biased relative the ground-based measurements within 1 km of the surface: low on 25 April (Fig. 9) and high on 27 April (Fig. 10), with closest agreement on 1 May (Fig. 11).

Otherwise only CPC data are available (for the remaining three case studies), and in each 1-km layer above the surface layer the ratio of the number concentration of the larger aerosol mode to the smaller aerosol mode is set to the experiment-wide median as a function of height, and the total of both modes is set to the median concentration measured by CPC in the corresponding 1-km layers.

For layers above the maximum measurement altitude the aerosol are taken to be that in the highest layer for which measurements are available (from either UHSAS or CPC, depending on the case study). Resulting aerosol specifications for simulations of each case are archived as Supplement 1.

Owing to the relatively simple modal structure of the input aerosol profiles derived here, an estimated coarse mode could be appended using climatological data or other field measurements (e.g., Corr et al., 2016), but we do not attempt that here.

## 4 Evaluation of hydrometeor size distributions in 20 May case study simulations

We demonstrate use of derived aerosol input data in the 20 May case study. Our simulations of the case (Table 1) use the NASA Unified Weather Research and Forecasting (NU-WRF) model (Peters-Lidard et al., 2015), set up as described by Tao et al. (2016), with an innermost domain of 1-km horizontal grid spacing (Tao et al., 2016, their Fig. 2). We compare observed hydrometeor size distribution properties with those simulated using Morrison et al. (2009) two-moment microphysics with hail and fixed droplet number concentration of 250 mg$^{-1}$ (BASE) and with prognostic droplet number concentration using semi-prognostic aerosol initialized with the aerosol profile described above (AERO). In the AERO simulation, mode-wise aerosol plus droplet number concentrations are everywhere initialized to the aerosol input profiles and fixed to them at the outermost domain boundaries. During simulation time, mode-wise aerosol are consumed by droplet collision-coalescence and transported as in Fridlind et al. (2012, their 'DHARMA-2M' simulation) and Endo et al. (2015, their 'DHARMA BIN' simulations). Unknown aerosol source terms are neglected; all else being equal, this increases the difference between BASE and AERO results (based on droplet number concentration differences discussed below). Since nucleation-mode aerosol (in the smallest fitted mode) are present very non-uniformly in the MC3E region, their concentration is set to zero in the BASE and





AERO simulations (cf. Fig. 8). In a sensitivity test simulation (NUCL), 8000 cm$^{-3}$ nucleation-mode particles are added to the bottom 2 km in a mode with geometric mean diameter of 0.005 $\mu$m and geometric standard deviation of 3, similar to the April case studies. We note that simulations use a preliminary version of the 20 May aerosol input data, which is negligibly different from the final version for the purposes of this study (the preliminary version is included in Supplement 1 for reference).

5    As noted above, the 20 May case is unique during MC3E owing to robust in situ sampling of extensive stratiform outflow from deep convection by the Citation aircraft (cf. Wang et al., 2015a; Wu and McFarquhar, 2016). Here we use ice number and mass size distributions derived from a PMS two-dimensional cloud (2DC) probe and a SPEC Inc. high-volume precipitation spectrometer (HVPS) probe (cf. Baumgardner et al., 2011, and references therein) on the aircraft. Since the derivation of number and mass size distributions and their integrals from such probes introduces substantial sources of uncertainty that are often not well quantified to date (e.g., Baumgardner et al., 2011), we use two independent derivations described in Wang et al. (2015a) and Wu and McFarquhar (2016), which differ in details of the methods used to process and estimate mass from the raw image probe data. For computation of mass median area-equivalent diameter (MMD$_{eq}$), we follow the Wu and McFarquhar (2016) approach for a first estimate, substituting the Baker and Lawson (2006) habit-independent mass-area dimensional relation for a second estimate (Table 2).

15    Over a 100x100-km domain centered on the C-SAPR radar, Fig. 12 shows the time series of surface precipitation derived from C-SAPR, from the National Mosaic and Multi-Sensor QPE system (Q2, Zhang et al., 2011) with and without rain gauge correction (Tang et al., 2014), and from the BASE simulation with fixed droplet number concentration in the region sampled by the Citation aircraft (region bounded by red rectangle in Fig. 13). Although the simulated squall line passes roughly an hour earlier than observed, we find qualitative agreement between predicted and retrieved maximum precipitation rates (about 20–30 mm h$^{-1}$) and the duration of rates greater than 50% of maximum (about 1 h). At a time representative of Citation aircraft sampling of the stratiform outflow (13:40 UTC), Fig. 13 shows a map of Q2 precipitation over the inner domain; the region sampled by the Citation aircraft is bounded by a red rectangle. Also shown is surface precipitation from the BASE simulation at the time of heavy stratiform precipitation (13:00 UTC, cf. Fig. 12). The BASE simulation shows a precipitation structure oriented in a band from southwest to northeast, similar morphologically to that observed (as do all subsequent simulations; not shown).

    Over the red-enclosed regions shown in Fig. 13, which bound the aircraft in situ sampling of stratiform conditions, Fig. 14 shows ice water content (IWC) and ice number concentration ($N_i$) derived from merged aircraft instruments alongside simulated values. Observed ice number concentrations at three well-sampled elevations (Table 2) are within the range of those reported from nine storms measured over Colorado and Oklahoma in May and June 2012 during the Deep Convective Clouds and Chemistry Experiment (Corr et al., 2016, 10–120 L$^{-1}$). We have omitted analysis of observations at lower elevations (temperatures warmer than $-10°$C) owing to initially suspected encounters with supercooled water, which can be difficult to confidently rule out (Wang et al., 2015a). Conditions at or near ice saturation are generally expected over heavy stratiform rain (e.g., Grim et al., 2009), but conditions ranging from liquid saturation to ice subsaturation above the stratiform melting layer could be associated with differing midlevel inflow positions or embedded convective-scale perturbations (e.g., Barnes and Houze, 2016). Later analyses of the 20 May case provide evidence of local ice subsaturation above the melting level associated



with bright band variability observed in C-SAPR fields (Kumjian et al., 2016). Here we focus on the top three elevations that were well-sampled and consistently more than 1 km above the variable bright band zone.

The aircraft observations shown in Table 2 are taken from five level legs flown between 13.9 and 14.9 UTC, except roughly one-third of the observations at $-23°C$ that are taken from an isolated level leg later in the same flight (cf. Wang et al.,
2015a, their Fig. 5). Since results are not sensitive to excluding the later samples, we consider the observations statistically representative of the 13.9–14.9 UTC time period. For our comparisons, simulations are sampled roughly one hour earlier, consistent with earlier squall line passage, using 10-minute output fields between 13 and 14 UTC.

With increasing elevation in the BASE run, summing all model ice classes, simulated IWC generally decreases while $N_i$ increases; both observational analyses show similar patterns in some respects, although the trend in $N_i$ across the three best
sampled elevations is not discernible. Overall, the largest apparent deviation of simulations from observations in Fig. 14 is roughly 4–10 times fewer ice crystals, although sampling remains relatively sparse and observational uncertainty could be very large. We do not attempt to quantify uncertainty in $N_i$ here owing to the difficulty of doing so and the relative lack of importance to analyses below, which are primarily focused on the size distributions of mass rather than number. In similar simulations of the 20 May case, Fan et al. (2015) show a similar order of magnitude underestimate of measured $N_i$.

Figures 15–17 show the underlying mass and number size distributions at the three well-sampled elevations (5.8, 6.7 and 7.6 km) as a function of ice crystal randomly oriented maximum dimension ($D_{max}$, cf. Wu and McFarquhar, 2016). At 5.8 km ($-10°C$), simulated and observed mass and number size distributions are compared in four mass concentration intervals spanning 0.2–1 g m$^{-3}$ (Fig. 15). The $D_{max}$ where the BASE mass size distributions peak is roughly 3–5 times larger than observed, consistent with underestimated $N_i$ within each mass concentration range. The $D_{max}$ where the BASE mass size distribution
peaks increases monotonically with increasing mass whereas the observed mass size distributions tend to peak consistently at $D_{max}$ of roughly 1–2 mm, generally independent of IWC range; other recent deep convection observations have found notably weak dependence of convective outflow ice size on mass concentration at fixed elevations (Fridlind et al., 2015; Leroy et al., 2015). We note that the $D_{max}$ at which observed and simulated size distribution lines cross one another (are equal) is greater for number than for mass because the effective density of the relevant ice particles (namely, snow) is less in the observations than
in the model microphysics scheme (0.1 g cm$^{-3}$, Morrison et al., 2009). Overall, there is a marked absence of particles with $D_{max} < 1000$ $\mu$m in the BASE simulation, suggesting that they are not produced or are lost via a process such as aggregation.

Observed number size distributions peak at $D_{max}$ of roughly 400 $\mu$m, which does not significantly change with either mass mixing ratio or elevation (cf. Figs. 16–17). At 6.7 and 7.6 km ($-16$ and $-23°C$). However, mass size distributions appear to fall into two modes: one peaking at $D_{max} \sim 500$ $\mu$m (most apparent at the lowest mass mixing ratios) and a second peaking at
$D_{max} \sim 1$ mm. The $D_{max}$ where observed number size distributions peak (at all elevations and mass concentrations) is similar to that where the smaller-mode mass size distribution peaks. In the observations, evolution with decreasing height from alignment to non-alignment of the mass and number size distribution peaks (namely, a shift of the mass size distribution peak to larger sizes that is not accompanied by a shift of the number size distribution peak) is suggestive of aggregation that is adequate to increase mass median $D_{max}$ but insufficient to increase number median $D_{max}$, conceivably owing in part to greater sticking
efficiency among larger colliding particles.



Subjective inspection of ice crystal images generally shows that aggregates are more common at larger sizes and lower elevations, consistent with the possibility that aggregation may be largely responsible for the coherent trend in observed particle size with elevation. However, the general irregularity of the ice particles (Fig. 18) makes confidently distinguishing aggregates from non-aggregates far more difficult than in a case where dendrites are the dominant habit, for instance, and aggregate fraction

can be readily estimated for simulation evaluation (e.g., Avramov et al., 2011). In this case, aggregates appear present at the highest elevation sampled ($-23°$C), but it has been suggested that aggregation may be a negligible process at temperatures warmer than $-20°$C (e.g., Barnes and Houze, 2016) and we cannot rule out the possibility that aggregation is not a dominant determinant of size distribution trends seen here in observations between $-10$ and $-23°$C.

Figure 19 demonstrates the effect of replacing fixed droplet number concentration in the BASE simulation with the aerosol

input data derived in Section 3 and prognostic droplet number concentration. The AERO ice size distributions are found to be largely unaffected compared with the BASE simulation. If nucleation-mode aerosols are added to the aerosol input file (NUCL simulation), results are similarly little affected. Inner-domain averages of cloud water mixing ratio and number concentration indicate that AERO droplet number concentrations are substantially smaller than fixed BASE values, especially aloft, and nucleation-mode aerosols are scarcely activated in the NUCL simulation (Fig. 20). A sensitivity test in which all

heterogeneous freezing parameterizations and ice multiplication mechanisms are turned off (HOMF), by contrast, results in substantially larger and fewer outflow ice crystals, worsening agreement with observations (cf. Fig. 19). Thus, we find that the combined effect of ice crystal formation parameterizations have a greater effect on outflow ice size than droplet spectra changes over the range in BASE versus AERO simulations. The fact that all of the simulations also substantially overestimate outflow ice size (where directly observed) is consistent with the possibility that the microphysics scheme could be missing

some critical aspects of ice nucleation or ice multiplication.

In all simulations $N_i$ decreases by roughly a factor of 8 between 7.6 and 5.8 km (as in Fig. 14). Observed $N_i$ does not show a discernible trend over the well-sampled elevations examined here (Table 2). These results suggest that simulated aggregation is more aggressive than observed in this case. In Fan et al. (2015) simulations of the same case with another two-moment scheme and a size-resolved microphysics scheme, $N_i$ decreases by roughly a factor of 20 over a similar altitude range (cf. their

Fig. 11b). Profiles of stratiform $N_i$ measured during the Bow Echo and Mesoscale Convective Vortex Experiment (BAMEX) exhibited 25% decline per degree C between 0 and $-10°$C, but were not reported at colder temperatures (McFarquhar et al., 2007; Smith et al., 2009). Because measurement uncertainty in $N_i$ remains essentially unquantified to date (e.g., Fridlind et al., 2007, uncertainty estimated at a factor of five), we do not attempt to draw conclusions at this point.

Radar reflectivity time series from the NEXRAD KVNX radar can place the aircraft-sampled elevations and locations into

greater context. By identifying columns of enhanced specific differential phase above the melting level in KVNX data, which can be taken as an indication of updraft location (van Lier-Walqui et al., 2016a), and using the nearest radiosonde to represent horizontal winds, we estimate that roughly two hours passed between ice detrainment from updrafts at roughly 35.5°N and Citation sampling at roughly 36.5°N (Fig. 21, left panels). A similar analysis of supercooled liquid water above the melting level and horizontal winds in the BASE simulation indicates a slightly shorter time period (Fig. 21, right panels); we have not





attempted to objectively optimize this analysis since results are not strongly sensitive to choice of time and location owing to the fact that conditions are quite horizontally uniform in both observed and simulated stratiform outflow.

Figure 22, derived from the fields shown in Fig. 21, illustrates that simulated reflectivity profiles below roughly 9 km are biased high in the AERO simulation (consistent with stratiform ice that is too large), but simulated reflectivity above roughly 10 km is biased low. Referring back to Fig. 21 (bottom panels) it is apparent that ice detrained from updraft tops in the BASE simulation may be either insufficiently concentrated or too small, but we have no direct measurements to confirm either possibility. Fan et al. (2015) simulations using both two-moment and size-resolved microphysics schemes show similar significant overestimates of 8-km reflectivity within stratiform outflow (cf. their Fig. 3b), indicative of similar biases in ice size (systematically larger than observed).

Figure 23 shows the median and inner half of raindrop mass-weighted mean diameter ($D_m$; the fourth moment of the drop number size distribution divided by the third moment) as retrieved from KVNX data following Ryzhkov et al. (2014), with an estimated uncertainty of roughly 5–10% (Thurai et al., 2012). The retrievals shown are made along the lowest-elevation KVNX beam, which varies in height with distance, but simulated values vary relatively little over that height range for the subregion selected to match the Citation sampling location. In that stratiform region (rectangular regions in Fig. 13) at the onset of the heaviest stratiform precipitation (13 UTC observed, 12 UTC simulated, cf. Fig. 12), simulated median $D_m$ is roughly 40% (0.7 mm) larger than observed, consistent with simulated stratiform ice size larger than observed at 5.8 km (roughly 2 km above the melting level).

Retrieved $D_m$ of 1.5–2 mm in the stratiform regime is on the high end of climatological values for various locations (cf. Thurai et al., 2010, their Fig. 2), but quite similar to stratiform values measured by disdrometer and retrieved from profiling radar in the same storm (cf. Williams, 2016, their Fig. 5b) and also in a tropical mesoscale convective system (cf. Varble et al., 2014b, their Fig. 17). Simulated $D_m$ values are larger than the upper end of stratiform values climatologically and show a high bias also found in similar simulations under tropical conditions using the same scheme (cf. Varble et al., 2014b, 'WRF-2M' in their Fig. 17).

Figure 24 shows simulated (BASE and AERO) and retrieved $D_m$ values as a function of collocated precipitation rate. Simulated stratiform rain $D_m$ values shown in Fig. 23 (selected to match the Citation location during aircraft sampling) are roughly equal to the microphysics scheme's breakup equilibrium value of 2.4 mm (cf. Morrison and Milbrandt, 2015), which is seen throughout the high precipitation rate limit in simulations. Observed $D_m$ asymptotes more monotonically to a relatively broader range in the high precipitation rate limit, where many retrieved values lie within retrieval uncertainty of 2.4 mm. We note that the existence, size distribution characteristics, and prevalence of breakup equilibrium has been elusive in nature, despite the idea that it might be guaranteed at much higher rain rates than found under stratiform conditions (e.g., McFarquhar, 2010; D'Adderio et al., 2015).

We also note that a mass-weighted mean diameter of 2.4 mm corresponds to a mean volume diameter of 1.1 mm for an exponential size distribution in the microphysics literature (e.g., Morrison and Milbrandt, 2015, their Appendix C), whereas the two diameters with the definition of the latter are used interchangeably in the radar literature (e.g., Testud et al., 2001). Considering raindrop size in general terms, the reduced droplet number concentrations in the AERO versus BASE simulation





are associated here with a reduction in the frequency of $D_m$ values below 2.4 mm at convective rain rates of 20–40 $\mu$m (cf. Fig. 24). This reduction is consistent with a pattern of increasing raindrop size with increasing aerosol or droplet number concentration shown in past modeling studies over a wide range of thermodynamic conditions (e.g., Storer et al., 2010) and also found over multi-day statistics using similar retrievals of raindrop size alongside ground-based aerosol observations under tropical conditions (May et al., 2011).

## 5 Summary and discussion

We report hygroscopic aerosol size distribution profiles for six convection case studies observed during the MC3E field campaign over Oklahoma. Each profile is derived by merging ground- and aircraft-based measurements. Missing aircraft data owing to instrument failures are filled by using experiment-wide analysis of flights where all instruments functioned well. The aerosol profiles, archived as Supplement 1, are intended for use in modeling studies of convection microphysics, where both aerosol and hydrometeor size distribution data are required to evaluate fidelity of model physics.

We demonstrate use of the aerosol size distribution profiles in NU-WRF simulations of the 20 May case study, where widespread stratiform outflow was also well sampled by aircraft. Using Morrison et al. (2009) two-moment microphysics with hail in NU-WRF as an illustrative example, we compare simulated ice size distributions with measurements made in the outflow region. Across several sensitivity tests (Table 1), we find that predicted and observed stratiform ice size distributions are similarly coherent within the stratiform region. However, simulated ice number concentrations ($N_i$) are roughly 5–10 times lesser than observed and the peak of ice mass size distributions roughly 3–5 times larger, correspondingly.

Across three well-sampled elevations between 5 and 8 km (at $-10$, $-17$, and $-23$°C), observed ice number size distributions peak at a randomly oriented maximum dimension ($D_{\max}$) of roughly 400 $\mu$m at all elevations, and lack a discernible vertical trend in total $N_i$ (Table 1). At the highest elevation sampled, the derived mass size distribution appears to peak at a $D_{\max}$ only slightly larger than 400 $\mu$m. At lower elevations, the peak $D_{\max}$ of the observed mass size distribution is shifted to a size twice as large, at roughly 1 mm, perhaps owing to aggregation that is apparent in ice crystal images. However, some mass remains in the smaller size range where numbers are concentrated.

In general, stratiform microphysics features seen in this 20 May mid-latitude squall line case appear notably similar to those observed in the tropics, as during the recent High Altitude Ice Crystals/High Ice Water Content (HAIC/HIWC) campaign that sought to robustly characterize ice properties that might be encountered by commercial aircraft transiting mesoscale convective systems around Darwin, Australia (Dezitter et al., 2013; Leroy et al., 2015, 2016). Perhaps most prominently, ice mass median area-equivalent diameter (MMD$_{\mathrm{eq}}$) values of 500–700 $\mu$m between $-15$ and $-25$°C (Table 2) are close to those found around Darwin in the same temperature range, and MMD$_{\mathrm{eq}}$ maxima of 900–1200 $\mu$m are also within the range found there (Leroy et al., 2016, their Figs. 6 and 9). Figs. 16 and 17, where the mass size distributions shown are visually integrable, show that the majority of mass in the 20 May case is generally found in a size range roughly bounded by half and twice the mass median size. Despite quite a bit of scatter, this condition found during HAIC/HIWC (Leroy et al., 2016, their Fig. 9) is indicative of a relatively narrow mode of ice mass around its median size, similar to that previously reported by Heymsfield (2003) from a



combination of tropical and mid-latitude measurements. We leave more detailed comparison of MC3E and HIWC/HAIC size distributions to future work, but here briefly note several other general similarities.

Although we have not identified the capped column habit that is common among convective ice crystal habits in the tropics (e.g., Grandin et al., 2014; Ackerman et al., 2015), there is a predominance of irregular, compact crystals on 20 May (cf.

Fig. 18), similar to those seen in tropical convective outflow during HAIC/HIWC (Leroy et al., 2015) and during the Tropical Composition, Cloud and Climate Coupling and NASA African Monsoon Multidisciplinary Analyses field campaigns (Lawson et al., 2010). A less prominent similarity that can be generally gleaned from Figs. 15 and 16 is that the ice size distributions on 20 May show relatively weak correlation of ice mass median $D_{\mathrm{max}}$ with IWC at fixed elevations aloft, especially in constrast to simulations here; a similar observation-simulation contrast has been reported under tropical conditions (Ackerman et al.,

2015, their Fig. 3). Over ten-degree temperature intervals colder than $-5°C$ (analogous to level flight legs here), HAIC/HIWC Darwin observations show a pattern of $MMD_{\mathrm{eq}}$ increasing or decreasing by less than 100–200 $\mu$m over a wider range of IWC sampled during HAIC/HIWC (up to $\sim3$ g m$^{-3}$ in Leroy et al., 2016) than sampled here (up to 1 g m$^{-3}$, Table 2). Profiles of Rayleigh reflectivity and Doppler velocity from a widespread tropical stratiform rain sampled during the Tropical Warm Pool—International Cloud Experiment (TWP-ICE) (Fridlind et al., 2015, their Fig. 11) also appear similar to the 20 May

observations (Fig. 22, Doppler velocity not shown here), consistent with generally similar stratiform ice size distributions over tropical and 20 May conditions.

In parcel simulations designed to study how relatively narrow mass size distributions of substantial outflow ice could develop within tropical updrfts detraining at roughly $-40°C$, Ackerman et al. (2015) concluded that copious crystal production at temperatures warmer than roughly $-10°C$ is required. In that study, copious mass concentrated in a relatively narrow mass

size distribution centered on an area-equivalent diameter of $\sim300$ $\mu$m required an ice growth time period much longer than that associated with homogeneous droplet freezing at 5 or so degrees warmer. Given an updraft profile, increasing number concentrations of ice at temperatures circa $-10°C$ increased the IWC carried to $-40°C$; any microphysical processes that competed with vapor growth of the ice crystals nucleated near $-10°C$ served to reduce the IWC available for detrainment at $-40°C$. Conversely, an absence of ice production near $-10°C$ favored microphysical pathways that produced larger hydrometeors that

sedimented from updrafts rather than detraining, consistent with simulations of tropical deep convection generally producing too little IWC over stratiform rain areas (e.g., Varble et al., 2014b).

We speculate that similar ice microphysical pathways that determine stratiform outflow ice properties are active in this 20 May case as in the tropical convection observed in many flights during HAIC/HIWC. This can be considered quite surprising since mid-latitude continental convection updrafts are well known to be much stronger than their tropical oceanic counterparts

(e.g., Liu and Zipser, 2015). However, it appears that deep convection updrafts may be direct source regions for individual outflow ice crystals (especially at upper elevations), consistent with the standard conceptual model of stratiform ice generation (e.g., Biggerstaff and Houze, 1991), and that ice which becomes stratiform rain may also exhibit rather narrow mass size distributions of relatively small crystals, consistent with an earlier and less complete data set gathered by Airbus (Grandin et al., 2014; Fridlind et al., 2015). The outflow ice size distributions (especially at lower elevations) are modified at least in

part by aggregation, consistent with layered patterns of ice crystal morphology obtained from dual-polarimetric radar particle





identification within tropical stratiform precipitation decks (Barnes and Houze, 2016); however, contributions to the structure of aged anvil ice size from differences in detrained size with elevation are not clear at temperatures between circa $-10$ and $-20°$C in the 20 May case, where signatures of dendritic growth are absent but reflectivity and mean Doppler velocity are generally increasing towards the melting level. In other words, the relative roles of detrained size, differential sedimentation, and aggregation in shaping vertical trends in stratiform ice size distribution are not clear.

The aircraft engine issues that motivated the HAIC/HIWC campaign are thought to be associated with unexpectedly high IWC for given radar reflectivities (Lawson et al., 1998; Mason and Grzych, 2011; Leroy et al., 2015). Such conditions, which require mass concentrated relatively narrowly in relatively small ice crystals, have been documented at mid-latitudes (Mason and Grzych, 2011). Whether or not they occurred in the 20 May case, it appears likely that a similar set of microphysical processes were active. Furthermore, it appears likely that such processes are not well represented in bin or bulk microphysics schemes generally (e.g., Ackerman et al., 2015; Fan et al., 2015; Varble et al., 2014b; Barnes and Houze, 2016). In one observation-driven modeling study, for instance, Zeng et al. (2011) propose an ad hoc "ice enhancement factor in the tropics" to bring simulations into statistical agreement with space-borne radar measurements. Developing tropical cumulus updrafts have also exhibited rapid ice production via ice multiplication that could depend on splinters formed during drop freezing rather than riming, which is not well understood to date and not represented in any commonly used microphysics scheme, and which may have a dominant impact on observed and simulated updraft glaciation rates (Lawson et al., 2015).

Differences between the simulated 20 May stratiform ice microphysics and observations shown here could arise variously from differences between model and natural ice crystal physical properties (density or structure of crystals), their associated fall speeds, aggregation and vapor growth rates, and the coupling of processes within outflow-generating updrafts, in addition to the ice crystal production rates via primary nucleation and ice multiplication. These factors require dedicated efforts to examine, but appear amenable to progress. For instance, in a follow-on study of this 20 May case (van Lier-Walqui et al., 2016b), we examine the stratiform column processes in isolation using a one-dimensional modeling approach to make a statistical determination of ice crystal properties and aggregation sticking efficiencies; for that work, the KAZR Doppler spectra are found to be essentially free from turbulence broadening in the quiescent stratiform environment, yielding copious information on ice size distribution variation over large regions of stratiform outflow. If outflow ice size distributions aloft are as similar to those present in detraining updrafts as suggested by HAIC/HIWC data from Darwin (at least for ice not sedimented rapidly within updrafts and prior to any substantial aggregation in the outflow), then the 20 May case study is also well suited to study of updraft microphysics.

Based on combined analysis of S-band (NEXRAD) and C-band dual-polarimetric radar signatures over several sites and seasons, it has been noted that the 20 May stratiform ice precipitation lacked the positive differential radar reflectivity commonly found in mid-latitude stratiform precipitation containing plate-like and oriented crystals (Williams et al., 2015). Williams et al. (2015) report a general absence of robust positive differential reflectivity in the trailing stratiform regions of "vigorous summer squall lines" and attribute that speculatively to the combined effects of irregular ice crystals and stronger electric fields. Strong electric fields have been associated with chain aggregates (e.g., Connolly et al., 2005), which to our knowledge were not profuse over the heavy stratiform rain region in the 20 May case. However, compact and irregular crystals and aggregates





are consistent with the available particle images, suggesting that lack of differential reflectivity signature may be indicative of a common stratiform microphysics regime across tropical mesoscale convective systems and mid-latitude summer squall lines.

Analysis of dual-polarimetric radar signatures from C-SAPR and KVNX using the quasi-vertical profile technique during stratiform rain on 20 May (Kumjian et al., 2016; Ryzhkov et al., 2016) have yielded conclusions generally consistent with the ice properties and microphysical pathways discussed. High specific differential phase in the absence of differential reflectivity enhancements in the elevation range examined here are consistent with relatively high ice number concentrations and the associated propensity for an active aggregation process despite an absence of dendritic growth. A strong negative gradient in differential reflectivity with elevation above the melting layer is indicative of efficient aggregation; we note that this is most intense approaching the melting level. However, the gradient changes sign near the uppermost elevations sampled by aircraft and examined here (cf. Kumjian et al., 2016, their Fig. 4), so we do not interpret this as conclusive evidence that aggregation is the primary process dominating the ice size distribution shape evolution colder than $-10°C$. Finally, within the melting layer, very high differential reflectivity and anomalously high backscatter differential phase are another indication of efficient aggregation above the melting layer (Trömel et al., 2014; Ryzhkov et al., 2016), confirmed by in situ observation of aggregates with $D_{max}$ greater than 17 mm just above it (not shown).

Such analyses of dual-polarimetric radar observations could be further systematically employed to identify the environmental conditions associated with stratiform microphysics regimes, assuming some variety exists, as has been suggested by (Leroy et al., 2016). Owing to the leading importance of tropical convection to global circulation and climate phenomena (e.g., Moncrieff et al., 2012), the dominant microphysics regime seen so far in HAIC/HIWC and some past measurements (Leroy et al., 2016), similar to that in the 20 May case, could be among those most important to properly represent in climate models. Aerosol interactions with convection could also be strongly dependent on the microphysics pathways active within a regime.

This is not the first MC3E convection modeling study to conclude that ice microphysics is not yet well-represented across microphysics schemes (e.g., Pu and Lin, 2015). Stratiform outflow from deep convection has also been previously identified as an area where different microphysics schemes in cloud-resolving or convection-permitting simulations produce particularly diverse results (e.g., Morrison et al., 2012; Varble et al., 2014b), with substantial associated impacts on simulated radiative fluxes (e.g., Fridlind et al., 2012; Wang et al., 2015b). Soundly advancing understanding of aerosol effects on deep convection requires better establishing and successfully reproducing in simulations the primary microphysical pathways operating under various environmental conditions. Identifying regimes where similar and distinct microphysical conditions can be identified in observations could usefully advance understanding and model development.

## 6   Code availability

Aerosol analysis codes are available in Interactive Data Language on request. The ice size distributions reported by Wu and McFarquhar (2016) were processed using the University of Illinois Optical Array Probe Processing Software (UIOPS), which is open source software available from https://github.com/weiwu5/UIOPS. NU-WRF software is available from http://nuwrf.gsfc.nasa.gov.





## 7 Data availability

Reported aircraft data are available from the DOE ARM program field campaign archive (http://www.arm.gov/campaigns/mc3e) and the NASA Precipitation Measurement Mission Ground Validation program archive (https://pmm.nasa.gov/science/ground-validation). Based on the raw aircraft microphysical measurements (Delene and Poellot, 2013), the ice number and mass size

distributions derived as reported by Wang et al. (2015a) and Wu and McFarquhar (2016) are available on request. Ground-based aerosol data are available from the DOE ARM program instrument data stream archive (https://www.arm.gov/data/datastreams). The C-SAPR quantitative precipitation estimate is available from the DOE ARM program as an evaluation product (http://www.arm.gov/dat NEXRAD measurements are available from the US government archive (http://catalog.data.gov/dataset). Specific differential phase and drop size distribution parameters calculated from NEXRAD measurements are available on request. NU-WRF sim-

ulations are also available on request.

*Author contributions.* A. Fridlind prepared the aerosol data analysis. X. Li and D. Wu modified NU-WRF to use derived input aerosol specifications, with the assistance of A. Ackerman. D. Wu ran NU-WRF simulations and compared results with in situ microphysics and precipitation observations. M. van Lier-Walqui prepared analysis of observed and simulated radar reflectivity, wind fields, and rain size distribution parameters. G. McFarquhar, W. Wu, X. Dong, and J. Wang provided airborne cloud microphysics measurements. A. Ryzhkov

and P. Zhang provided rain size distribution parameter retrievals. M. Poellot, A. Neumann, and J. Tomlinson provided airborne aerosol measurements. A. Fridlind and W.-K. Tao coordinated this project with companion studies.

*Competing interests.* The authors declare that they have no conflict of interest.

*Acknowledgements.* This work was supported by the Office of Science (BER), U.S. Department of Energy, under agreements DE-SC0006988 and DE-SC0014605, and by the NASA Radiation Sciences Program. All measurements were obtained from the Atmospheric Radiation Mea-

surement (ARM) Program sponsored by the U.S. Department of Energy, Office of Science, Office of Biological and Environmental Research, Climate and Environmental Sciences Division. Operation of the University of North Dakota Citation aircraft was supported through NASA Grant NNX10AN38G. NU-WRF is supported by the NASA Modeling, Analysis and Prediction (MAP) program. Resources supporting this work were provided by the NASA High-End Computing (HEC) Program through the NASA Advanced Supercomputing (NAS) Division at Ames Research Center and NASA's Center for Climate Simulation (NCCS) at Goddard Space Flight Center. We thank Tami Toto and

Andrew Vogelmann for supplying their algorithm for fitting multiple lognormal modes to a measured aerosol number size distribution.





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





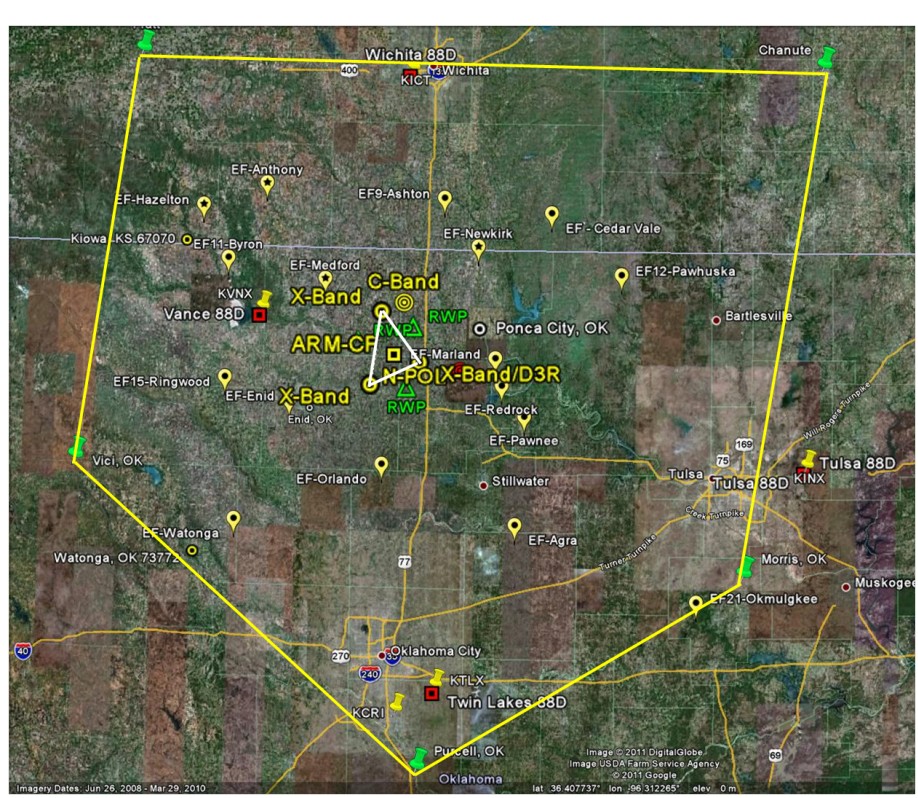

**Figure 1.** MC3E operations map: around the ARM central facility (small yellow square) are arrayed the X-band radars (white triangle), the sounding array (large yellow pentagon), and the C-SAPR radar (yellow bull's eye symbol). Figure courtesy of Michael Jensen.





**Figure 2.** KAZR radar reflectivity at the central facility during six case studies that begin on 25 and 27 April, and 1, 20, 23, and 24 May 2011.



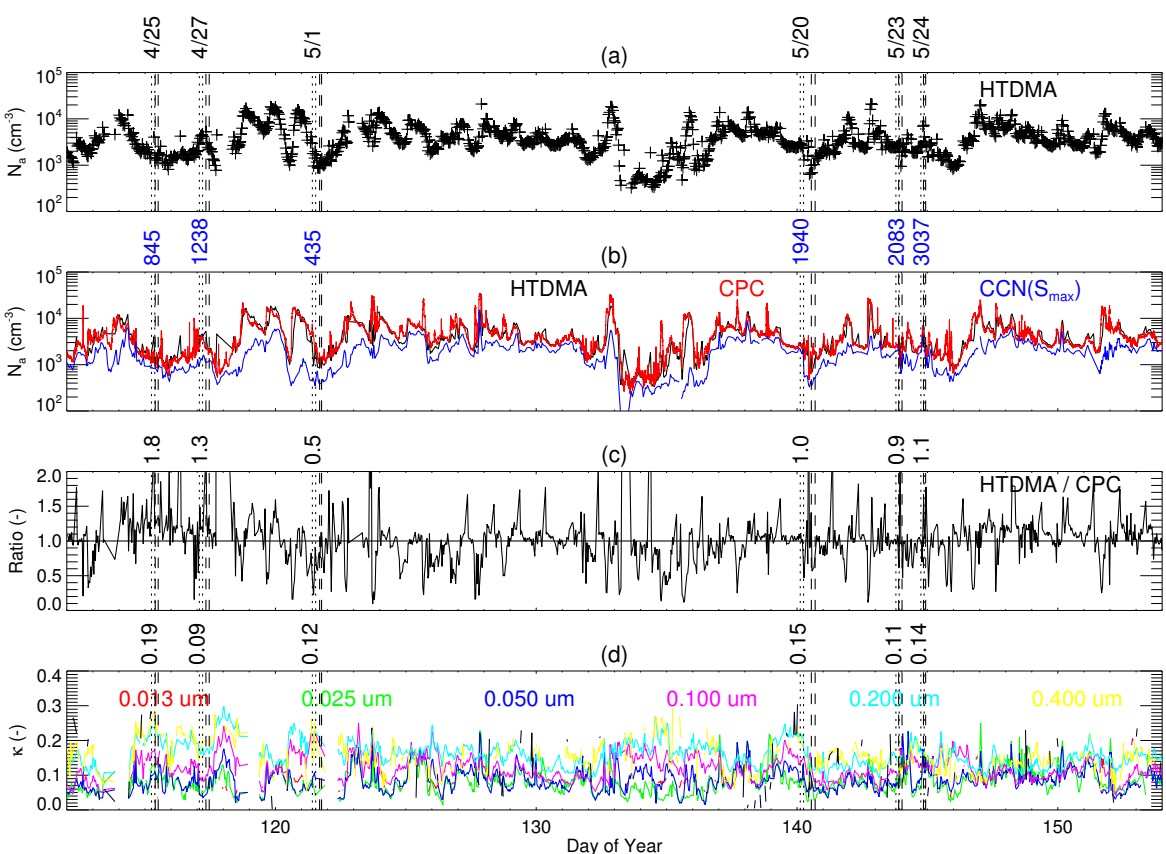

**Figure 3.** Ground-based aerosol properties at the central facility during MC3E: (a) total number concentration ($N_a$) measured by the HT-DMA; (b) $N_a$ measured by the HTDMA, CPC, and CCN at maximum supersaturation circa 1%; (c) ratio of $N_a$ measured by HTDMA to that measured by CPC; and (d) hygroscopicity parameter ($\kappa$) measured by the HTDMA at six sizes. For each case study, pairs of dashed and dotted vertical lines bound the Citation flight duration and a two-hour pre-rain period, respectively; values above plots are pre-rain period averages, linearly averaged over available sizes for $\kappa$.




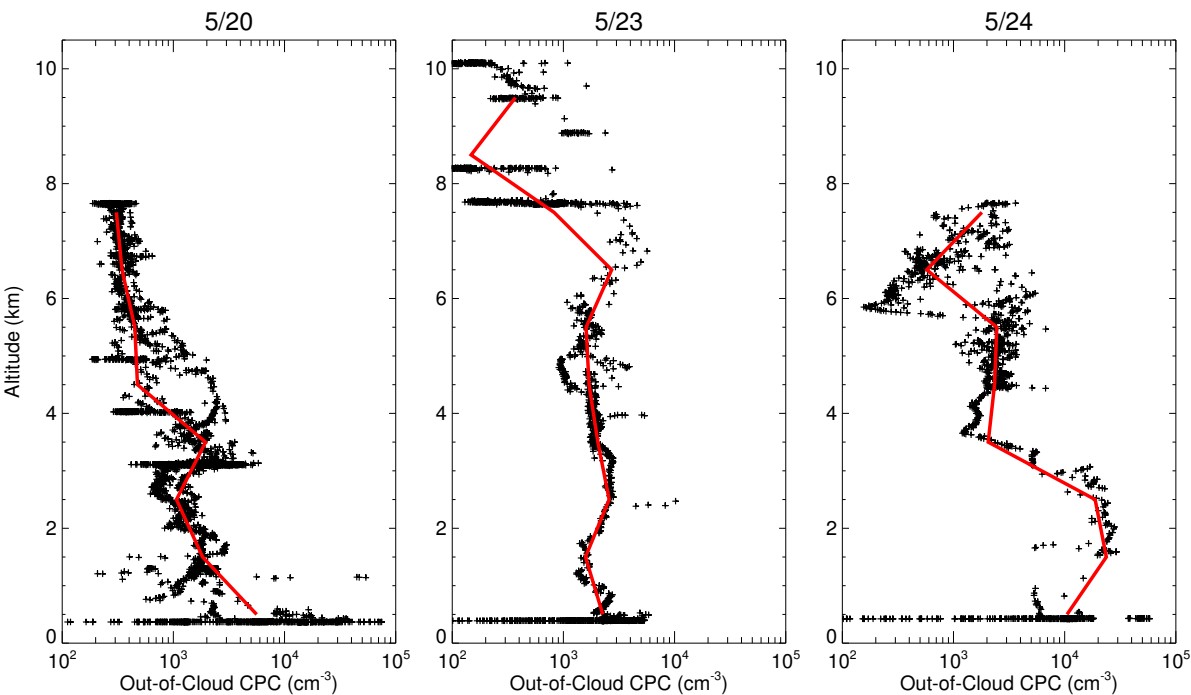

**Figure 4.** Airborne out-of-cloud CPC measurements of aerosol number concentration available on case study days (black symbols), and average profile over km-deep layers (red lines).





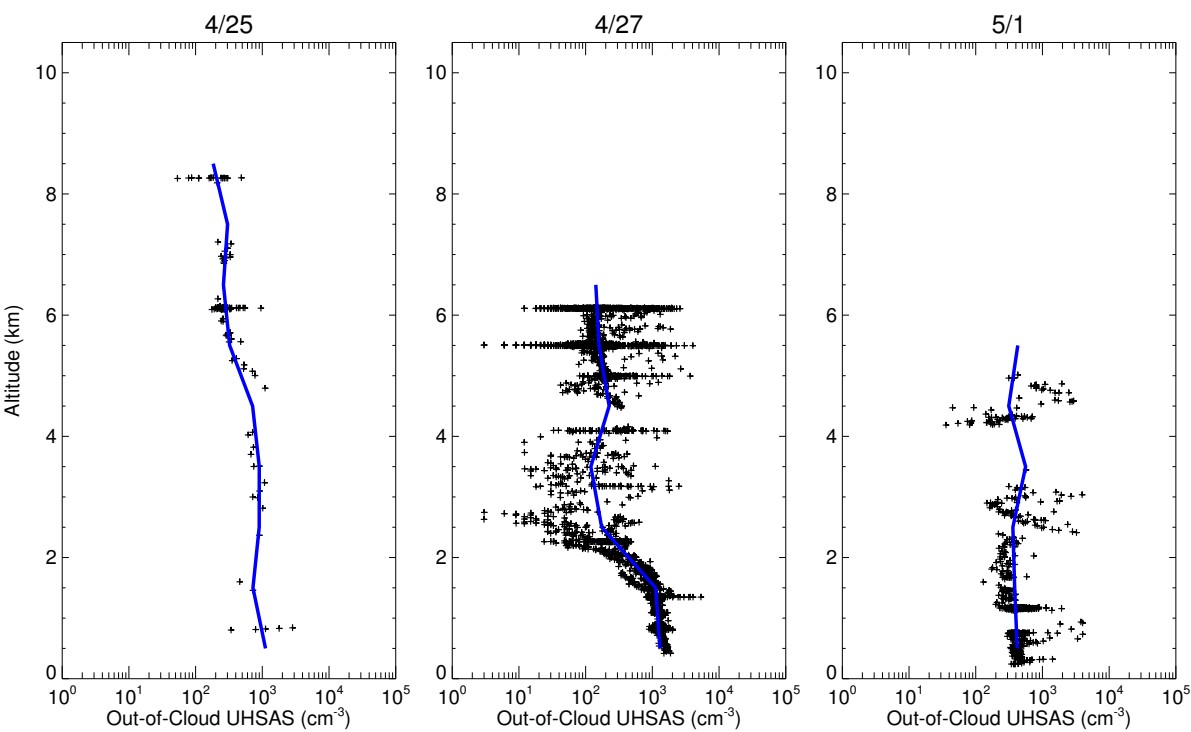

**Figure 5.** Airborne out-of-cloud UHSAS measurements of aerosol number concentration available on case study days (black symbols), and average profile over km-deep layers (blue lines).



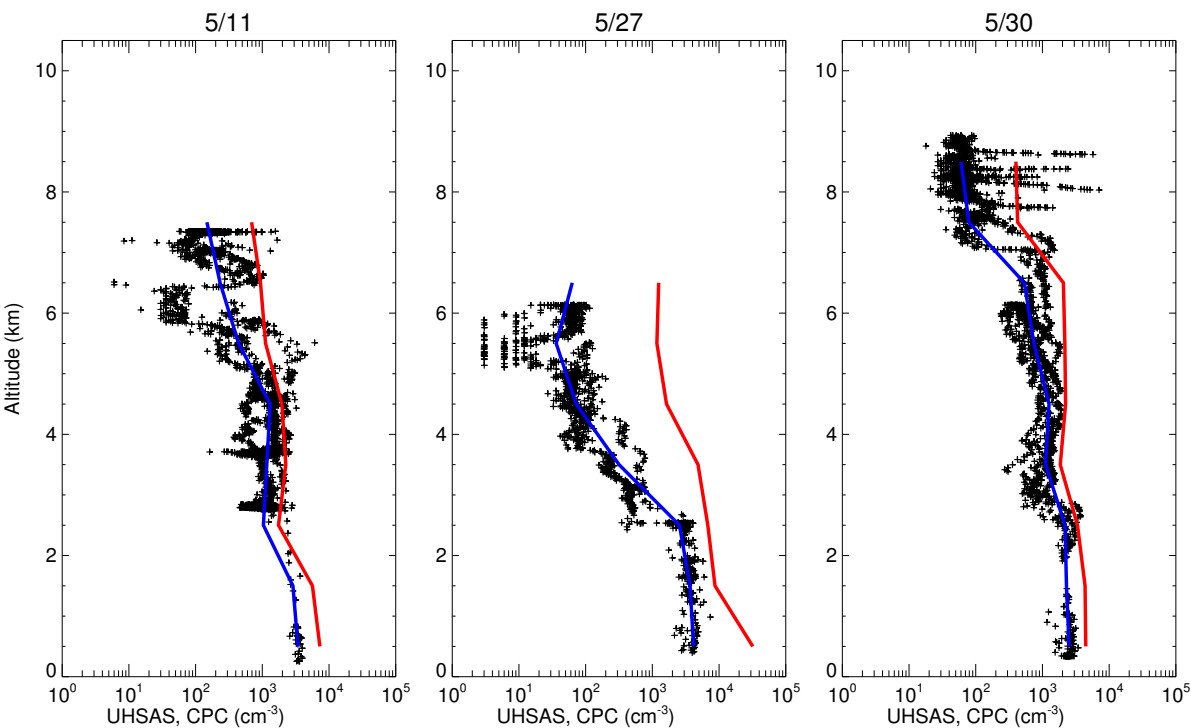

**Figure 6.** Airborne out-of-cloud UHSAS measurements of aerosol number concentration (black symbols) on three flight days with CPC measurements, and average UHSAS and CPC profiles over km-deep layers (blue and red lines, respectively).





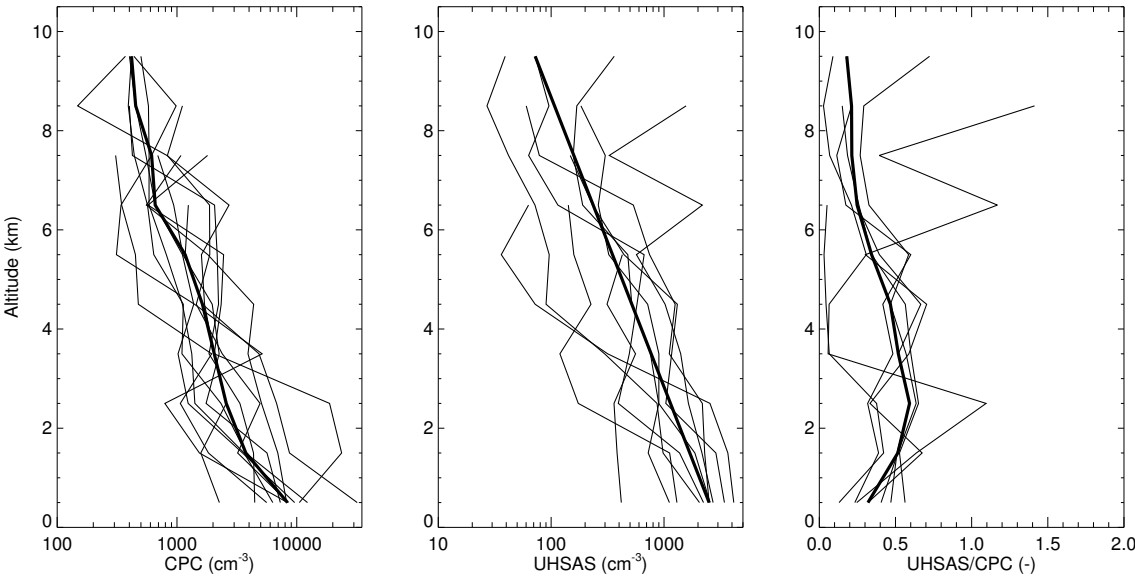

**Figure 7.** Average CPC and UHSAS aerosol number concentration profiles over km-deep layers (red and blue lines, respectively), the ratio of UHSAS to CPC concentrations on flights with both instruments (thin black lines), and a profile of layer-wise median ratio (thick black line).







**Figure 8.** Aerosol dry size distributions ($dN_a/dlogD_a$) during a two-hour pre-rain period for each case study: measurements from the HTDMA during the pre-rain period (colored solid lines), lognormal fits to measurements (colored dashed lines), and a number-weighted average (black dashed line, see text). For each measurement time are listed the fitted mode-wise number concentration, geometric mean dry diameter and standard deviation. Also listed for the pre-rain period average is mode-wise number-weighted average hygrscopicity parameter ($\kappa$). For the 20 May case also shown are distributions with zero and 8000 cm$^{-3}$ particles in the nucleation mode (dotted lines).





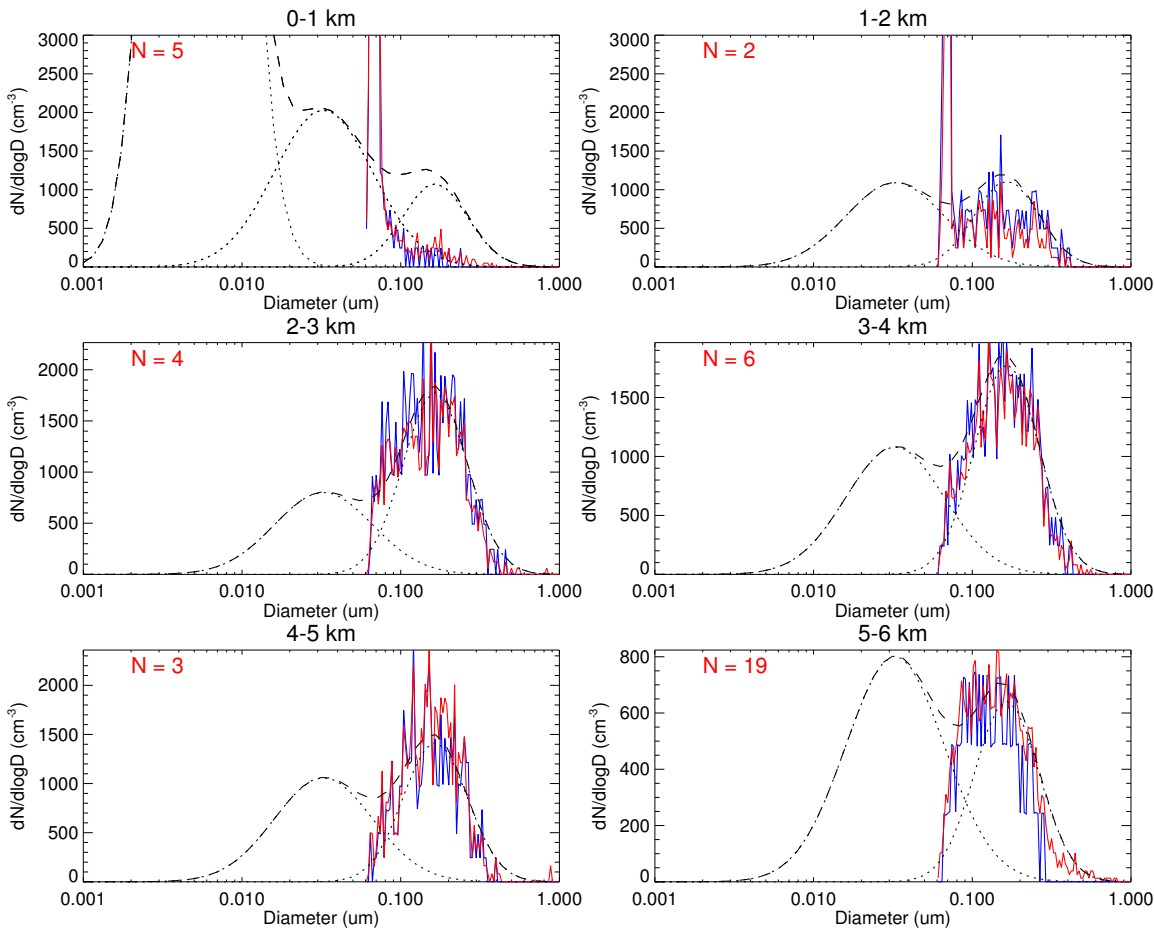

**Figure 9.** Derived mode-wise and total aerosol number size distributions over km-deep layers (black dotted and dashed lines, respectively) compared with bin-wise mean and median out-of-cloud UHSAS size distributions (red and blue lines, respectively) for the 25 April case study.





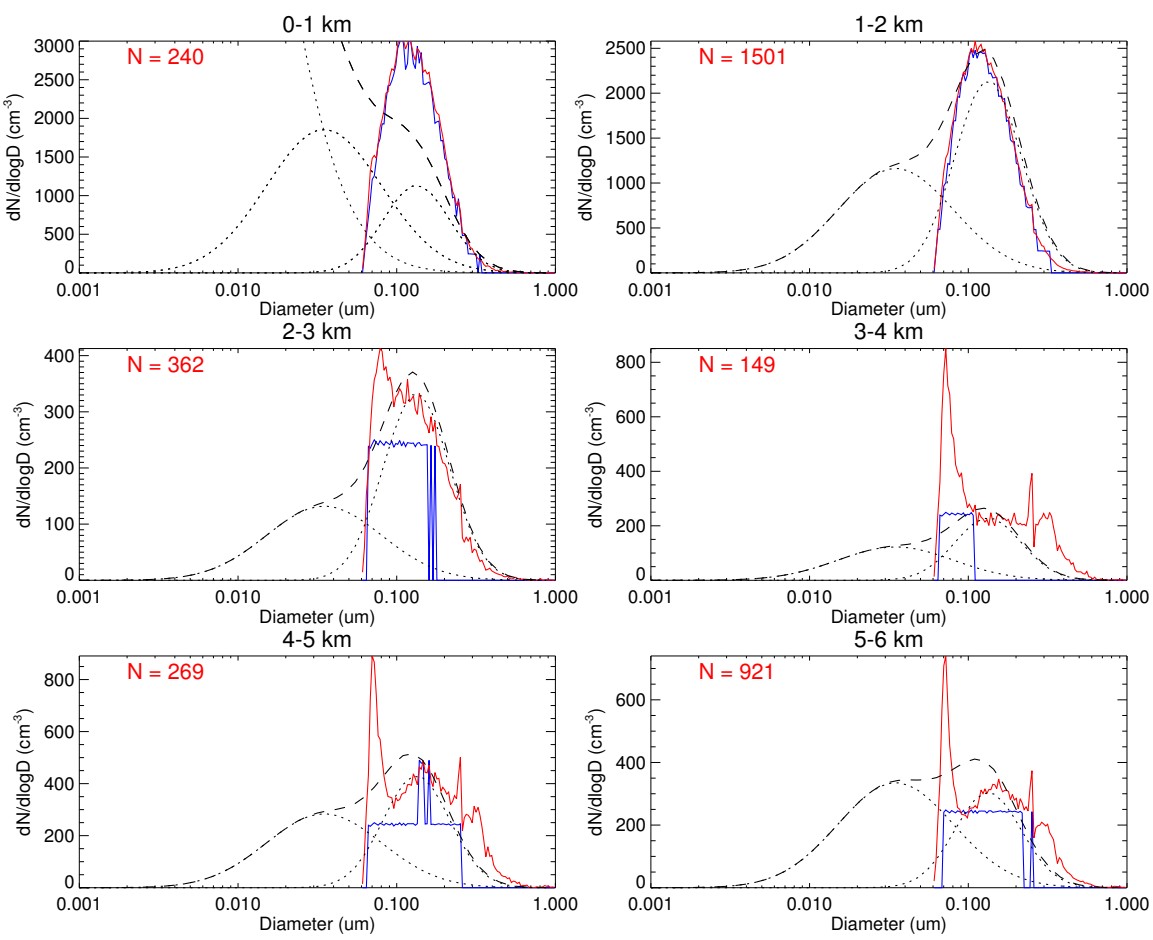

**Figure 10.** As in Fig. 9 for the 27 April case study.



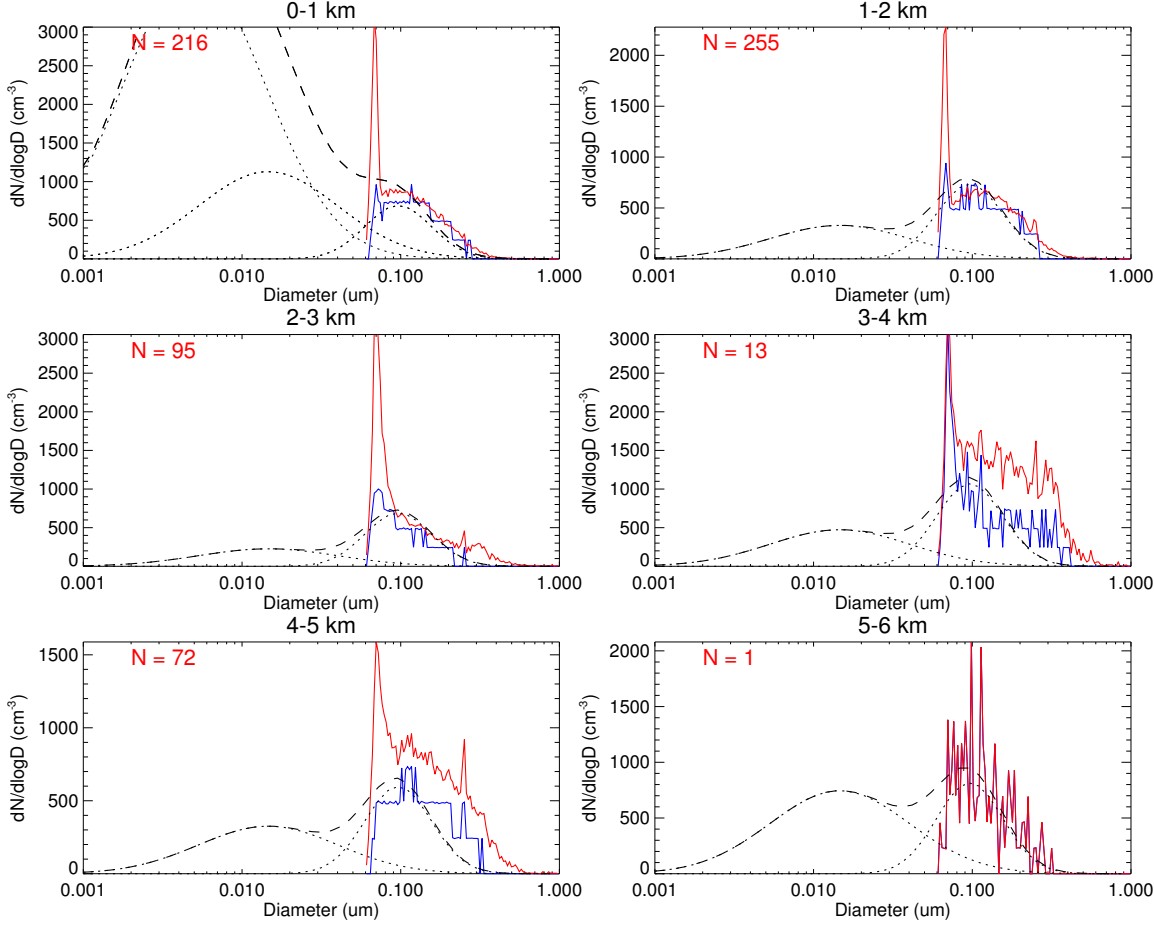

**Figure 11.** As in Fig. 9 for the 1 May case study.

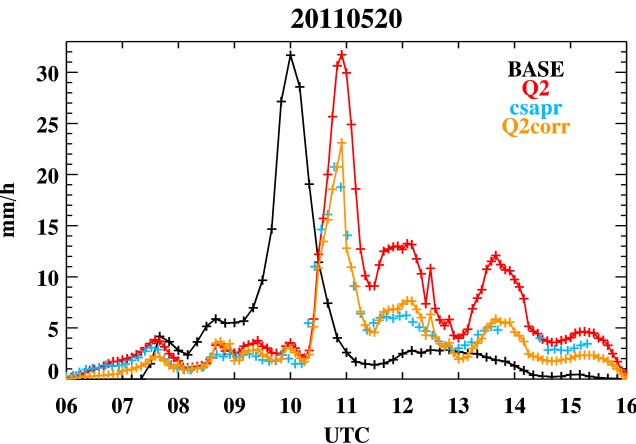

**Figure 12.** Time series of precipitation from the BASE simulation, Q2, C-SAPR, and gauge-corrected Q2 averaged over the region bounded by a red rectangle in Fig. 13.





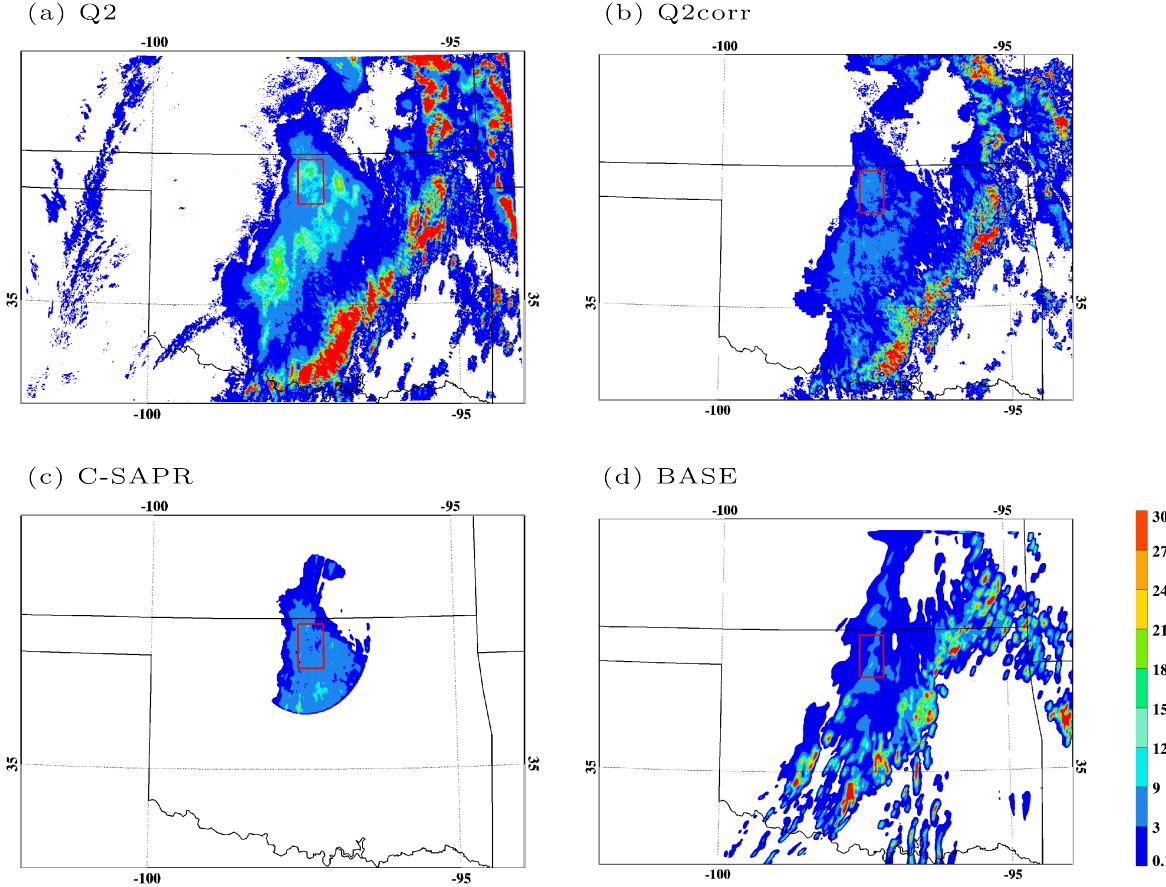

**Figure 13.** Surface precipitation rate (mm h$^{-1}$) from Q2 at 14:00 UTC (upper left), gauge-corrected Q2 (upper right, see text), from C-SAPR at 13:40 UTC (lower left), and in the BASE simulation at 13:00 UTC (lower right). Red rectangles bound the Citation aircraft flight legs examined here.



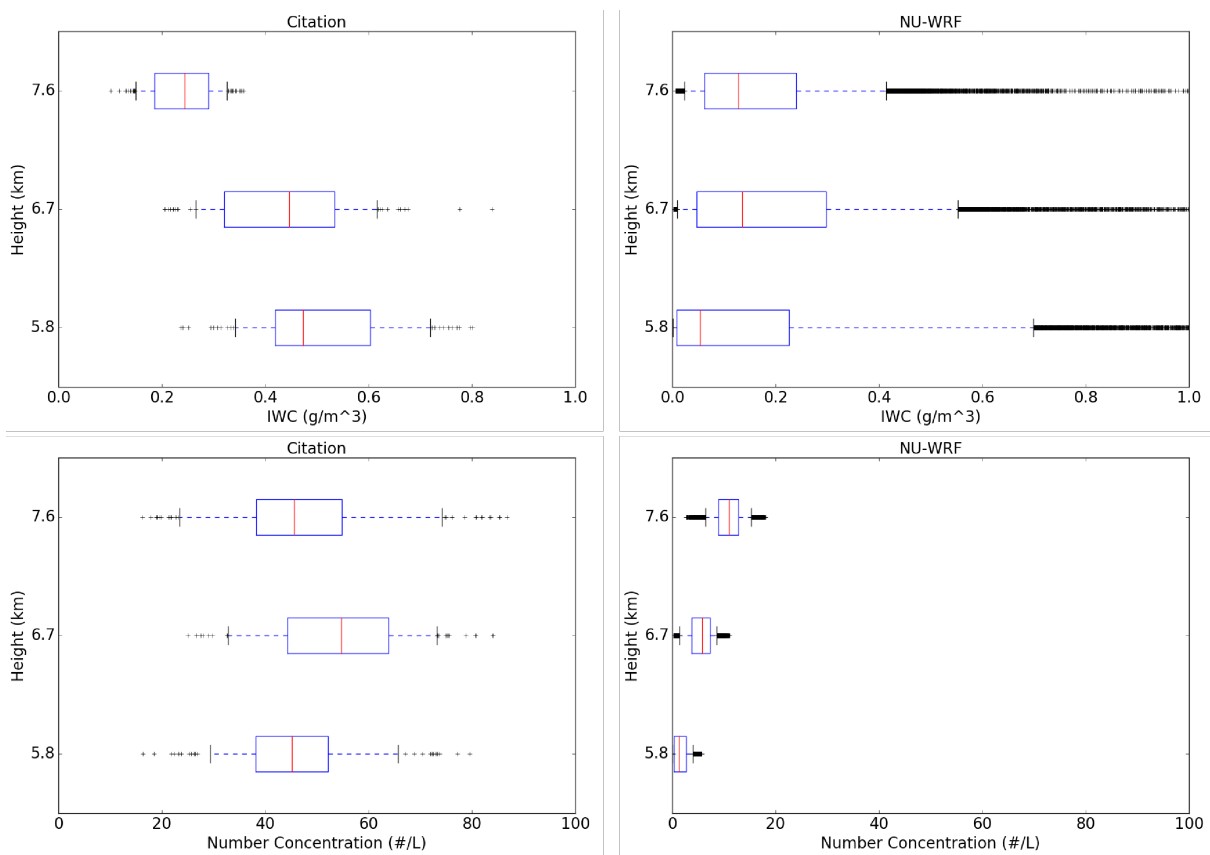

**Figure 14.** Total ice water content (IWC, top) and ice number concentration ($N_i$, bottom) derived from aircraft observations (left), as reported by Wang et al. (2015a) and Wu and McFarquhar (2016), and from the BASE simulation (right), within the respective red-bounded geographic regions shown in Fig. 13. Simulated ice is the sum of all ice classes. Observed ice is the sum of all size bins shown in Fig. 15. Box and whisker symbols represent the median, inner half, and 5th and 95th percentiles.





**Figure 15.** Ice mass size distributions (left) and number size distributions (right) in four ranges of ice water content (IWC, ranges given in parentheses in units of g m$^{-3}$) as derived from Wang et al. (2015a, red) and Wu and McFarquhar (2016, blue) and from the BASE simulation (black) at 5.8 km ($-10°$C) within the respective red-bounded geographic regions shown in Fig. 13. Error bars indicate one standard deviation of values sampled at each size. Simulated ice is the sum of all ice classes at each size. The simulated ice bin size is sphere diameter calculated from the bulk density of each ice class. The measured ice size is the randomly oriented maximum dimension.





**Figure 16.** As in Fig. 15 except at 6.7 km ($-16°$C). No IWC greater than 0.8 g m$^{-3}$ was measured at 6.7 km.





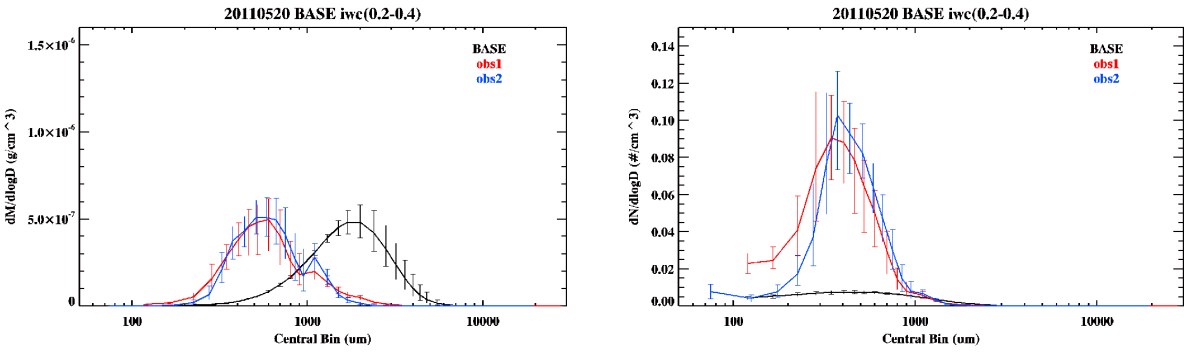

**Figure 17.** As in Fig. 15 except at 7.6 km ($-23°$C). No IWC greater than 0.4 g m$^{-3}$ was measured at 7.6 km.



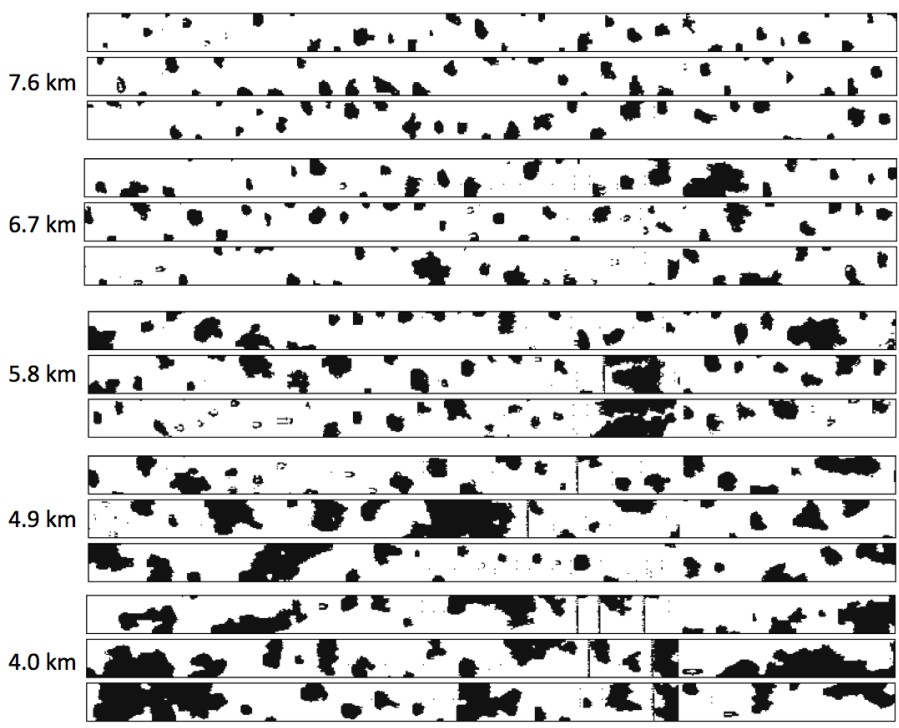

**Figure 18.** 2DC image collage from flight legs above the melting level in the 20 May stratiform outflow region. Three time series examples are given at each elevation. The vertical dimension of each time series is 960 $\mu$m. Here we focus on the top three elevations that are greater than ~1 km above the variable melting level height of ~3.9 km (see text).







**Figure 19.** Simulated ice number and mass size distributions averaged over the red-bounded geographic region shown in Fig. 13 over 13–14 UTC. Each panel shows the results from four simulations as labeled (see Table 1).





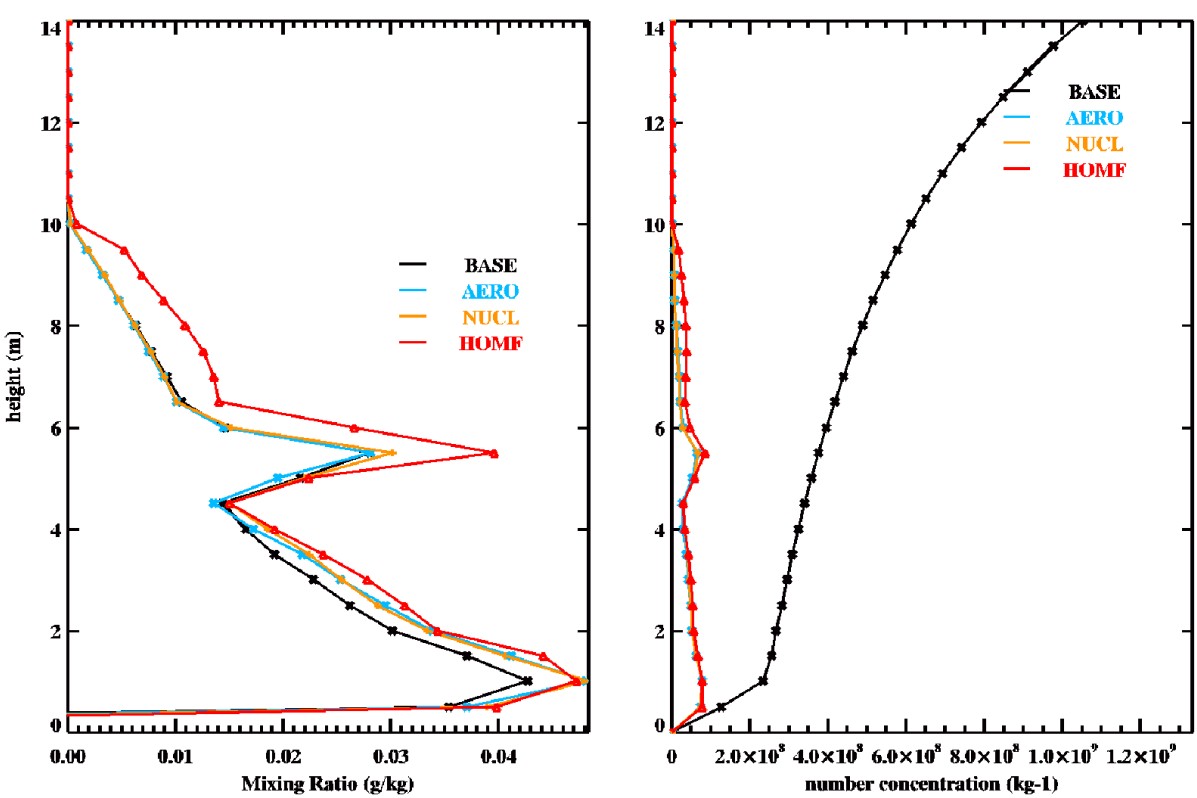

**Figure 20.** Simulated cloud droplet mixing ratios (left) and number concentrations (right) averaged over the full domain shown in Fig. 13 over 13–14 UTC.





**Figure 21.** From KVNX observations (left) and AERO simulation (right): identification of a typical updraft object in units of dBZ-km (top, see text), movement of objects with mean winds to intersection with the aircraft sampling location projected on 2-km radar reflectivity in units of dBZ (middle) and reflectivity time series obtained from column-wise averages over encircled domains at each time step from time of updraft identification (bottom, see text; vertical solid and dashed lines correspond to mid-point of hour-long averages shown in Fig. 22).





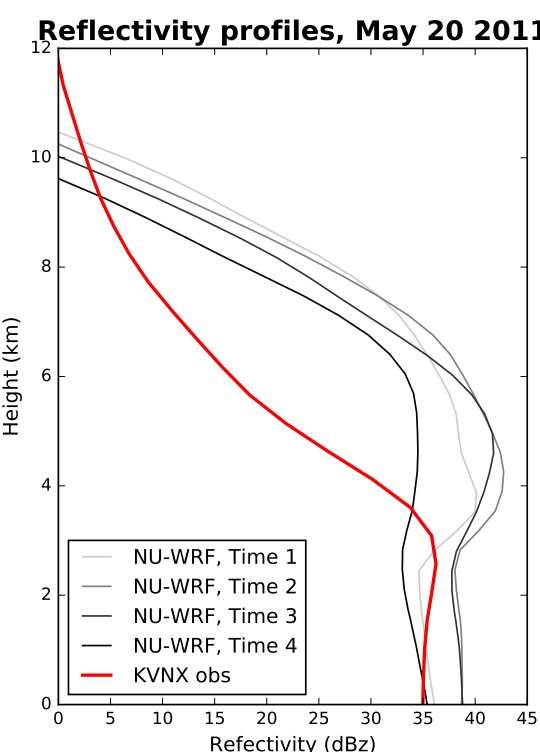

**Figure 22.** Reflectivity profiles obtained from one-hour average of reflectivity time series shown in Fig. 21 from KVNX (red line) and AERO simulation (light to dark grey lines).





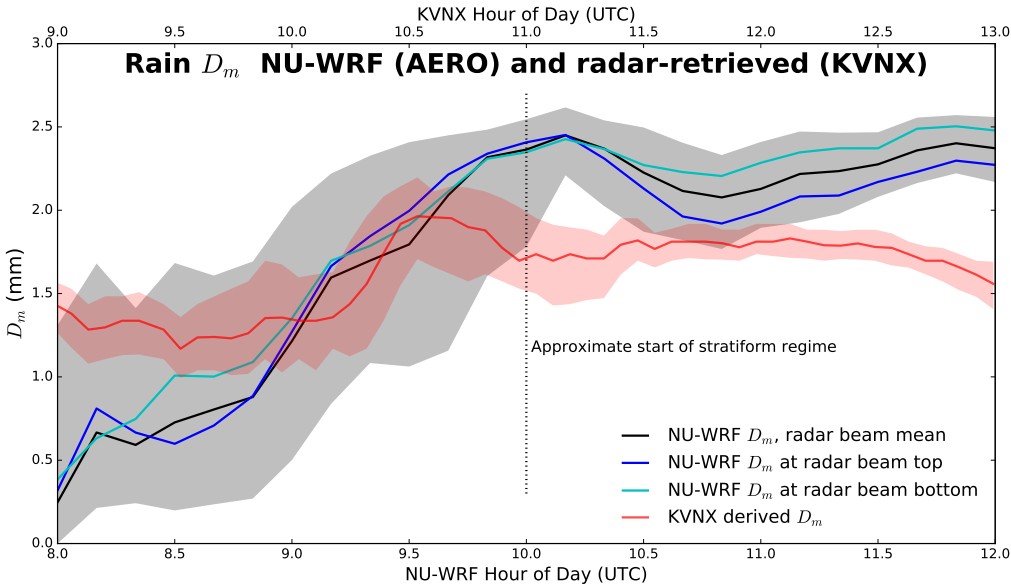

**Figure 23.** Mass-weighted mean diameter ($D_m$) as a function of time in the AERO simulation and in retrievals averaged over the respective red-bounded geographic regions shown in Fig. 13. Lines indicate median values (see legend). Shaded regions indicate inner half of retrieved values and simulated values at the radar beam mean height. Note offset in time axes (top and bottom) to align approximate timing in observations versus simulations.





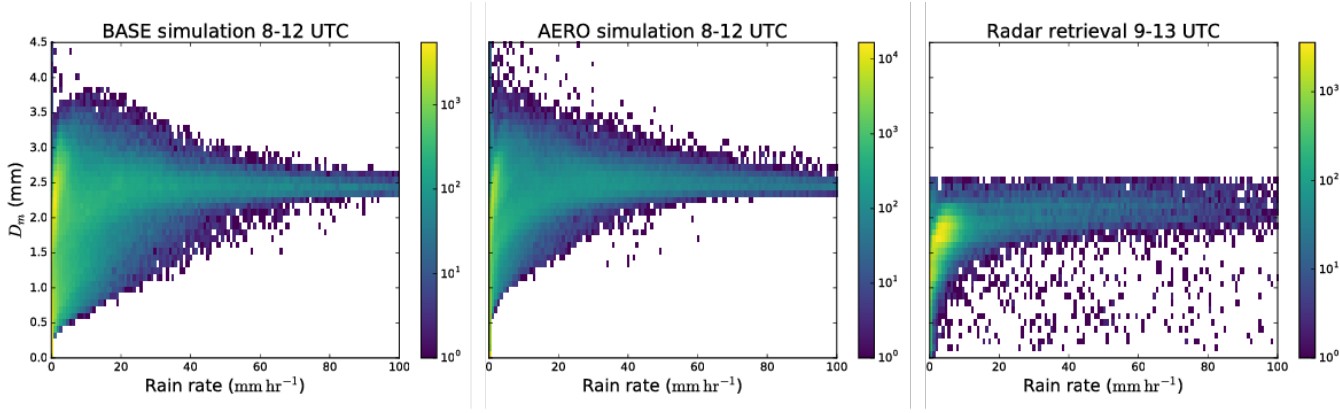

**Figure 24.** Joint histogram of mass-weighted mean diameter ($D_m$) and collocated precipitation rate in BASE and AERO simulations and retrievals averaged over the red-bounded geographic region shown in Fig. 13 over 8–12 UTC (simulated) or 9–13 UTC (retrieved).





**Table 1.** Summary of NU-WRF simulations without and with aerosol input profile and prognostic droplet number concentration ($N_d$).

| Experiment | Prognostic $N_d$ | Nucleation-mode aerosol | Homogeneous freezing only |
|---|---|---|---|
| BASE | — | — | — |
| AERO | ✓ | — | — |
| NUCL | ✓ | ✓ | — |
| HOMF | ✓ | — | ✓ |





**Table 2.** Aircraft-observed ice water content (IWC), ice crystal number concentration ($N_i$), and mass median area-equivalent diameter (MMD$_{eq}$) statistics by elevation, with range given over two derivation methods (see text).

| Elevation (km) | Temperature (C) | Mean IWC (g m$^{-3}$) | Max. IWC (g m$^{-3}$) | Mean $N_i$ (L$^{-1}$) | Max. $N_i$ (L$^{-1}$) | Mean MMD$_{eq}$ ($\mu$m) | Max. MMD$_{eq}$ ($\mu$m) |
|---|---|---|---|---|---|---|---|
| 7.6 km | −23 | 0.21–0.28 | 0.38–0.43 | 39–47 | 78-87 | 515–530 | 900–1025 |
| 6.7 km | −17 | 0.44–0.50 | 0.94–0.96 | 51–54 | 84–100 | 701–704 | 1025–1200 |
| 5.8 km | −10 | 0.52–0.56 | 0.89–1.0 | 45–46 | 72–80 | 948–993 | 1850–2200 |