# Peer review of "Use of an observation-based aerosol profile in simulations of a mid-latitude squall line during MC3E: Similarity of stratiform ice microphysics to tropical conditions"

_Atmospheric Chemistry and Physics, 2016_

## Referee Comment (RC1) · Anonymous Referee #1 · 1 Dec 2016

General comments This article constructs hygroscopic aerosol size distribution profiles from MC3E aircraft and ground-based data over six days. These profiles are used to run 4 NU-WRF simulations of a squall line case study. Observed and simulated cloud ice microphysical properties in the stratiform outflow region are then compared.

The work is very comprehensive and cites the existing literature thoroughly. The results about similarity between continental and tropical ice microphysics are quite interesting. Although factors like "fall speeds, aggregation and vapor growth rates, [etc]" are listed in the results, I would have appreciated more discussion on how the modeled

ice microphysics might be improved to bring something like the number and mass size distributions into better agreement with observations. I missed also a discussion of the one-hour offset between the simulated and observed rain event initiation. Is there a hypothesis for this?

There were two other points on which I would have appreciated clarification. I was surprised by the result that modeling only homogeneous freezing (HOMF) results in "substantially larger and fewer outflow ice crystals". Normally homogeneous freezing yields many more and smaller crystals (e.g. DeMott et al. 1998 GRL). Why does the opposite occur here? Then I found the results for the size distributions in Figures 14 to 17 and radar reflectivity in Figure 22 incongruous: the distribution comparisons indicate that the simulated ice crystals are far too big, while it is suggested from the reflectivity comparison that the simulated ice is too small. Am I missing something? Clarification in both cases would be helpful.

Otherwise my comments are related to readability. I find the article rather figure-heavy, and I think the results would be made be more accessible if the figures were condensed in some places and simplified in others. For example, Figure 2 is only referred to once, and since only the 20 May panel is particularly relevant, this panel could be combined with Figure 12. In Figure 13, only the rain gauge-corrected QPE measurements and BASE simulation are discussed, so panels a and c could be removed. Or Figures 17 and 18 could be moved to Supplemental Information, since the altitudinal dependence of Ni and mass distributions is already seen between Figures 15 and 16 and the discussion of 2DC images is quite brief.

I think breaking down the "Evaluation of hydrometeor size distributions in 20 May case study simulations" section into subsections, e.g. "Precipitation intensity", "Mass and number concentration distributions", and "Radar retrievals", would also ease readability.

Specific comments Page 4, Line 6 – Please be consistent in the instrument acronyms.

What is called the "DMA" here is later called the "HTDMA" in Figure 3 and introduced as the "TDMA" in Section 2. Again for the CPC, it is not always clear whether the measurements to which you refer are form the ground-based or aircraft CPC; it is inferred from the other instruments you mention. You could make this more explicit.

Page 6, Line 31 – The statement "unknown aerosol source terms are neglected" is unclear to me. The airport and power plants are mentioned in the section of Aerosol input data, but there is no discussion of back trajectories or systematic confirmation of hypothesized sources.

Page 7, Lines 1-2 – Is there also a quantitative basis (other than "similarity to April case studies") for the 8000 cm-3 and 0.005 um values chosen in the NUCL simulation? If so, this should be mentioned.

Page 7, Line 3 – The statement "simulations use a preliminary version of the 20 May aerosol input data" is unclear to me. The Aerosol input data section does not mention multiple processings or versions.

Page 8, Line 1 and Page 47, Table 2 – Could you please include the standard deviation in the "top three elevations", e.g. 7.6 $\pm$ x m, and associated temperatures?

Page 11, Line 16 – It is not clear what "similarly coherent" means here. Could you word this more substantively?

Page 28, Figure 7 – The caption indicates that the CPC profiles on the left and UHSAS profiles on the right are in red and blue respectively as in Figures 4 and 5, but this is not the case. The thick black line for layer-wise median ratio is not so easily distinguished from the thinner black lines; perhaps the UHSAS/CPC traces can also be changed from black in the rightmost subplot. Finally it is not clear what the "layer-wise" ratio means; are these values also calculated for km-deep layers?

Page 29, Figure 8 – The numbers in the subpanels of this figure need to be moved to a table. This will significantly ease comparing the values between days and allow the

y-axis to be readjusted for better comparison of the different traces. It is also unclear to me what the various colors (red, green, blue, purple, black) represent. The caption refers to "measurement time", but this should be clarified. A brief discussion of why the 2-mode fit is better than the 3-mode and vice versa at certain times might also be included in the third paragraph of page 5.

Page 35, Figure 14 – In my opinion, this figure could be removed, and the simulated values added to Table 2.

Page 41, Figure 20 – I am confused by the black BASE trace for number concentration. Doesn't this simulation have a fixed droplet concentration of 250 mg-1, as stated on page 6, line 26?

Page 42, Figure 21 – It is unclear whether only the top left panel is an integrated reflectivity; it seems so given its different scale, but this should be clarified in the caption. A definition of ZHH (as the horizontally-polarized radar reflectivity, right?), along with definitions for the pink, white, and red circles in various subpanels, would help in the interpretation of this figure.

Page 43, Figure 22 – "Time 1", "Time 2", etc. have not been defined for the simulations. It would be clearer to label the gray traces 'AERO, Time1' etc. so that the reader knows these are only from that simulation.

Technical comments / suggestions Page 3, Line 25 – A term like "droplet activation" or "ice nucleation" or "new particle formation" would more clearly indicate the process(es) meant by "aerosol consumption" here.

Page 3, Line 26 – Remove the second "be".

Page 6, Lines 9 – 10 – Add a 'to': "appears to be variably biased relative to the ground-based measurements".

Page 8, Lines 16-22 – Reword through here for clarity, e.g. "Consistent with under-estimated Ni, the Dmax at which BASE mass distributions peak is roughly 3-5 times

larger than that at which the observed distribution peaks. The Dmax at which the BASE mass distributions peak increases monotonically with increasing mass concentration, whereas the observed mass distributions tend to..

Page 8, Line 28 – There is an unfinished sentence beginning with "At 6.7 and 7.6 km".

Page 12, Line 18 – "Updrfts" to "updrafts"

Page 14, Line 4 – "have" to "has"

Page 14, Line 6 – "are" to "is"

Page 22, Figure 1 – It would ease readability if the ARM central facility were marked with a color other than yellow, since the pentagon, bull's eyes, and thumbtacks are all yellow as well.

Page 24, Figure 3, panel d – Is there a red trace for 0.013 um here? If so, it is not visible.

Page 33, Figure 12 – It would ease readability if Q2 were expanded to National Mosaic and Multi-Sensor QPE system in this caption, as well as in the text, and again if QPE were expanded here and in the text.

Pages 36-38, Figures 15-17 – The red and blue traces should be labeled PMS 2DC and HVPS rather than obs1 and obs2.

Page 41, Figure 20 – The y-axis should be '[km]' not '[m]'.

---

## Referee Comment (RC2) · Anonymous Referee #2 · 27 Dec 2016

This study examines and reports aerosol size distribution profiles for six convection case studies observed during the MC3E field campaign, intended for use in model simulation of those cases. The authors demonstrate use of the aerosol size distribution profiles in NU-WRF simulations of the 20 May case study with Morrison two-moment microphysics focusing on examining the stratiform cloud microphysical properties. There are some interesting findings such as ice crystal number concentrations are consistently dominated by a single mode near Dmax of 400 $\mu$m, and a mass mode near Dmax of 1000 $\mu$m becomes dominant with decreasing elevation to the $-10\ 0C$. Therefore, the study is worthy being published. However, this reviewer does have some

concerns about the current form as listed below,

(1) I am a little confused about the objectives for the second half of the paper that demonstrates the use of the derived aerosol size distribution. The Introduction does not have a clear statement about the goal of this part. Their results show that simulation using the aerosol size distribution derived does not much affect ice microphysics and stratiform microphysical properties including particle size distribution. These results kind of dispute the importance of aerosol size distribution used in model simulations. Logically, to show the importance of the developed product (i.e., aerosol size distribution), the paper should present results that are significantly changed by aerosol size distribution such as precipitation rate, convection, etc. But the authors did not go to this direction and kind of ignored the point about the importance of the derived aerosol size distribution to MCS simulations. This is ok only if the authors clearly state the reasons for doing what they chose to do and the relevant objectives.

(2) Section 3 does not have a clear structure. This part is very important to the entire paper, and the authors need to be clear about (a) the methodology of how the aerosol size distributions are derived, (b) the final products provided to the community, and (c) the discussion about caveats and uncertainties. However, the current writing in this section makes readers difficult to get those. The authors are still talking that the methodology in the last 4 paragraphs of this section.

(3) The contribution of small CCN to droplet nucleation and ice particle concentration at upper-levels needs some further examination. The conclusion is premature. See comment #20.

(4) About Section 5, although I enjoyed reading the discussion, much of the discussion should be moved to the Introduction since they are the very relevant literature studies providing the background for this work. In addition, some of the things discussed here are not even mentioned in the main text or not much related (for example, the lack of the positive differential radar reflectivity and the importance of the tropical convection

in global circulation).

(5) There are many inconsistencies between Figure, Figure captions, and the corresponding text, and also a few figure captions do not clearly describe the figures. There are quite a few sentences what do not make sense or are wrongly stated. Please refer to the specific comments below for the details.

(6) Too many figures: some figures can be combined such as Fig 4 and 5, and some are not key to the main points such as Fig. 9-11, and Fig. 16-17, which could be the options for the supplemental materials since there is already a supplemental file.

Detailed comments,

1. P1 Line 14-15, not sure what you want to say here, especially about the specific meaning of "the microphysics pathways associated with deep tropical convection outflow".

2. P2 Line 2, aerosol should be plural here.

3. P2 Line 10-14, this is a very long sentence. Suggest to break into two sentences to make it easier to read.

4. P2, last paragraph, the last a few sentences of this paragraph need to be revised to clearly state the objectives of this study. If the objective is to achieve more accurate simulations, then is the goal achieved?

5. P3 Line29, aerosol should be plural here.

6. P4, Line 5-11: the description here about Figure 3 suggests Na is from DMA or CPC and kappa is from HTDMA. However, the Figure 3 caption said only HTDMA, and no DMA data is shown. Please clarify the inconsistency. In addition, description about instrumental uncertainty for each instrument would be helpful here.

7. P4, Line 15-16, something is missing in the later half of the sentence. Otherwise, it does not make sense.

8. P4, Line 15-19, the description here would be clearer if the ratios of CCN to CPC aerosol concentrations are shown.

9. P5, Line 8 and Lin 17: what are non-case-study dates and case study dates?

10. P5, I do not understand what is said in the sentence "UHSAS/CPC again sometimes decrease, not because UHSAS decreases but because CPC increases, consistent with evidence that the surface is also a source of fine particles". CPC increases suggested more small particles, which could be from particle nucleation at the elevated altitudes. This is observed quite often. So, I do not understand why we can infer that surface is the source.

11. Figure 7, there are no red and blue lines.

12. Figure 8, why are there two colored solid lines for the measurement from HTDMA? It is really confusing with so many numbers on each panel and the description is not clear for some numbers such as the numbers at the right bottom part of each panel. Strongly suggest to use a table to show the parameters for the three modes. Also, need to explain the purpose of showing the 0 and 8000 cm-3 in the nucleation mode for May 20 case.

13. Fig. 8, there are such large differences in the measurements of HTDMA for 4/25 and 5/24 in the smallest mode (although it is not clear each colored solid line represent), then any fit should have very large uncertainty. Is it meaningful for such a fit?

14. Fig. 9, what is N? What is total aerosol number size distribution?

15. P6 Line 27-32, the text here is confusing: first, need to be specific about aerosol configurations in AERO. It is not enough to just say "initialized with the aerosol profile described above" since it is not clear "above". To me, Fig. 8 is above but there are many different aerosol parameters listed on the panel for 5/20. Second, since AERO has prognostic droplet number concentrations, I do not understand why need to fix droplet number concentrations at the boundary? Shouldn't fixing aerosol be enough?

Third, I do not understand "Unknown aerosol source terms are neglected", thus I am confused with the later part if the sentence "how all else being equal, this increases the difference between BASE and AERO results". Lastly, it is not clear what cloud microphysics scheme is used for other simulations besides BASE.

16. P6 Line 33, BASE should have no aerosol since droplet number is not prognostic as shown in Table 1.

17. P7 Line 1-2, why 8000 cm-3? This sounds a very large aerosol number concentration.

18. P7, the third paragraph and Fig. 12: Q2 and Q2corr cover the entire domain, why not compare the precipitation over the entire domain? Suggest to add such a plot to Fig. 12 (after all, it would be a more robust comparison compared with that over a small domain of 100x100 km2).

19. Figure 14, There is only one observation dataset shown in the figure, why are there two sources (Wang etal. 2015a and Wu and McFarquhar 2016)? The related discussion about the two measurements is on P8 Line 9 but the figure does not show both.

20. P9 Line 12-14, If Morrison scheme is used, do you consider second droplet nucleation or only cloud-base nucleation is considered? I would expect secondary nucleation at higher altitudes could make significant differences if small CCN is present. Therefore, I would suggest to do another test with the secondary nucleation considered if it is not considered in the NUCL.

21. P9 Line 18-20, I think the point is mainly supported by much smaller ice particle number concentration simulated by the model.

22. Figure 21, please define Zm and ZHH. Also, I do not understand why each panel is plotted for a different time? And the figure order does not reflect a time evolution, and the color legend is different for the same type of figures between observation and

model simulation such as Panels 2 and 3. What does the red color denote in the first four panels?

23. P10 Line 5-6, why suddenly talking about BASE since only AERO is compare with observations in both Figures 21 and 22.

24. P10 Line 30-31, suggest to reword the sentence. It is not easy to understand currently.

25. P11 Line 15-16, "we find that predicted and observed stratiform ice size distributions are similarly coherent within the stratiform region": I am not sure what this sentence really means since simulated and observed size distributions are totally different as shown in Figs. 14-17.

26. The third paragraph in Section 5: this paragraph summarizes observed results. It is natural to comparatively describe how model does here, and this information is missing from the summary currently.

---

## Author Comment (AC1) · 16 Feb 2017

General comments

This article constructs hygroscopic aerosol size distribution profiles from MC3E aircraft and ground-based data over six days. These profiles are used to run 4 NU-WRF simulations of a squall line case study. Observed and simulated cloud ice microphysical properties in the stratiform outflow region are then compared. The work is very comprehensive and cites the existing literature thoroughly. The results about similarity between continental and tropical ice microphysics are quite interesting.

We very much appreciate the helpful questions and comments. Point-by-point responses below have greatly improved the manuscript by reducing figures, adding section numbers, and making corrections and clarifications throughout.

Although factors like "fall speeds, aggregation and vapor growth rates, [etc]" are listed in the results, I would have appreciated more discussion on how the modeled ice microphysics might be improved to bring something like the number and mass size distributions into better agreement with observations.

Clarification added to Section 5: "The NU-WRF biases relative to observations shown here are consistent with the hypothesis that microphysics schemes are missing a key aspect of an updraft microphysics pathway that can largely determine outflow size, most likely associated with warm-temperature ice multiplication (e.g., Ackerman et al., 2015; Lawson et al., 2015; Ladino et al., 2017). Here we show that NU-WRF biases in stratiform ice mass size distribution are worsened when warm-temperature contributions to ice formation are decreased; Ackerman et al. (2015) find the same in parcel simulations and also demonstrate how biases can be decreased when warm-temperature contributions are substantially increased. In the simulations shown here, we also speculate that gravitational collection of stratiform ice may be too efficient, at least in the mid-troposphere, as evidenced by reflectivity increasing and number concentration decreasing substantially more rapidly than observed between 8 and 6 km (cf. Figs. 10 and 17)."

I missed also a discussion of the one-hour offset between the simulated and observed rain event initiation. Is there a hypothesis for this?

Clarification added to section 4.1: "The simulated squall line passes roughly an hour earlier than observed, which could be attributable to two general causes: (i) uncertainties in the initial and boundary conditions, including those influential to surface heat fluxes, and (ii) errors in model parameterization components, including microphysics scheme elements, which can independently influence the rainfall structure in NU-WRF simulations in this case (cf. Tao et al., 2016, their Fig. 11)."

There were two other points on which I would have appreciated clarification. I was surprised by the result that modeling only homogeneous freezing (HOMF) results in "substantially larger and

fewer outflow ice crystals". Normally homogeneous freezing yields many more and smaller crystals (e.g. DeMott et al. 1998 GRL). Why does the opposite occur here?

Clarification added to section 4.2.2: "Whereas favoring homogeneous freezing of droplets generally yields more ice particles in an updraft parcel (e.g., DeMott et al., 1998), here we find the opposite in aged stratiform outflow, where snow is the dominant hydrometeor class. Snow number concentration maxima intermittently reach ~500 $L^{-1}$ in all simulations except HOMF, where they reach only ~30 $L^{-1}$. Since 500 $L^{-1}$ is the limit imposed on the Cooper (1986) parameterization contributions to total ice number concentration (see Section 4.1), we conclude that removing that source is likely chiefly responsible for larger ice in HOMF outflow. We note that ice number concentrations are not conserved by design in order to enforce limits on size distribution slope parameters (Morrison et al., 2009), which complicates drawing firm conclusions about the contributions of specific processes."

Then I found the results for the size distributions in Figures 14 to 17 and radar reflectivity in Figure 22 incongruous: the distribution comparisons indicate that the simulated ice crystals are far too big, while it is suggested from the reflectivity comparison that the simulated ice is too small. Am I missing something? Clarification in both cases would be helpful.

Clarification added to section 4.3.1: "Thus, specifically at the elevations where the aircraft sampled (Fig. 16, white bars in observed reflectivity), simulated reflectivity is substantially greater than observed, consistent with ice particles substantially larger than observed (Figs. 11–13), but that is not the case at all elevations."

Otherwise my comments are related to readability. I find the article rather figure-heavy, and I think the results would be made be more accessible if the figures were condensed in some places and simplified in others. For example, Figure 2 is only referred to once, and since only the 20 May panel is particularly relevant, this panel could be combined with Figure 12. In Figure 13, only the rain gauge-corrected QPE measurements and BASE simulation are discussed, so panels a and c could be removed. Or Figures 17 and 18 could be moved to Supplemental Information, since the altitudinal dependence of Ni and mass distributions is already seen between Figures 15 and 16 and the discussion of 2DC images is quite brief.

We combined Figs. 4 and 5 and removed 6, 10–11, and 19–20. We retained 2 (for reader to quickly assess other case study conditions), 13 (emphasizes substantial uncertainty in rainfall products), 17 (15-17 are main focus), and 18 (for modelers to know what ice looks like).

I think breaking down the "Evaluation of hydrometeor size distributions in 20 May case study simulations" section into subsections, e.g. "Precipitation intensity", "Mass and number concentration distributions", and "Radar retrievals", would also ease readability.

We now use two levels of subsections in Sections 3 and 4.

Specific comments

Page 4, Line 6 – Please be consistent in the instrument acronyms. What is called the "DMA" here is later called the "HTDMA" in Figure 3 and introduced as the "TDMA" in Section 2. Again for the CPC, it is not always clear whether the measurements to which you refer are form the groundbased or aircraft CPC; it is inferred from the other instruments you mention. You could make this more explicit.

HTDMA now used throughout. CPC now always preceded by "ground-based" or "airborne."

Page 6, Line 31 – The statement "unknown aerosol source terms are neglected" is unclear to me. The airport and power plants are mentioned in the section of Aerosol input data, but there is no discussion of back trajectories or systematic confirmation of hypothesized sources.

By unknown we meant that aerosol source terms cannot be readily observed and specified. Simplification and clarification made: "Aerosol source terms beyond advection across outer domain boundaries are neglected (e.g., primary emission and gas-to-particle conversion)."

Page 7, Lines 1-2 – Is there also a quantitative basis (other than "similarity to April case studies") for the 8000 cm-3 and 0.005 um values chosen in the NUCL simulation? If so, this should be mentioned.

Clarification added also in response to referee 2: "Based on the April and 1 May nucleation-mode fits listed in Fig. 6, this represents the most commonly fit mode diameter and mode standard deviation, and a modest number concentration (maximum on 1 May) that is lower than typically observed in the 10–30-nm diameter range during intense new particle formation events (e.g., Crippa and Pryor, 2013)."

Page 7, Line 3 – The statement "simulations use a preliminary version of the 20 May aerosol input data" is unclear to me. The Aerosol input data section does not mention multiple processings or versions.

Clarification added: "During the course of this study, minor changes were made to aerosol observation processing concurrently with the simulations being run; simulations therefore use a preliminary version of the 20 May aerosol input data, which is negligibly different from the final version for our purposes. AERO and NUCL aerosol input files are included in Supplement 1 for completeness."

Page 8, Line 1 and Page 47, Table 2 – Could you please include the standard deviation in the "top three elevations", e.g. 7.6 ± x m, and associated temperatures?

We prefer not to complicate the table because the elevations and temperatures are in a narrow range based on level legs within horizontally homogeneous conditions and the table is already complicated by showing a range of minimum and maximum values from two observational data sets.

Page 11, Line 16 – It is not clear what "similarly coherent" means here. Could you word this more substantively?

Reworded to "both predicted and observed stratiform ice size distributions exhibit relatively well-defined properties that do not vary rapidly in time."

Page 28, Figure 7 – The caption indicates that the CPC profiles on the left and UHSAS profiles on the right are in red and blue respectively as in Figures 4 and 5, but this is not the case. The thick black line for layer-wise median ratio is not so easily distinguished from the thinner black lines; perhaps the UHSAS/CPC traces can also be changed from black in the rightmost subplot. Finally

it is not clear what the "layer-wise" ratio means; are these values also calculated for km-deep layers?

Clarification added to figure and caption: "The median of airborne CPC and UHSAS aerosol number concentrations within 1-km-deep layers for each MC3E flight, and the ratio of those median values for the seven flights with both instruments (black lines). The median of profile values at each elevation (red lines) are archived as Supplement 2."

Page 29, Figure 8 – The numbers in the subpanels of this figure need to be moved to a table. This will significantly ease comparing the values between days and allow the y-axis to be readjusted for better comparison of the different traces. It is also unclear to me what the various colors (red, green, blue, purple, black) represent. The caption refers to "measurement time", but this should be clarified. A brief discussion of why the 2-mode fit is better than the 3-mode and vice versa at certain times might also be included in the third paragraph of page 5.

We used a fixed vertical axis to emphasize case study differences. The black values are archived with Supplement 1 and we disagree that the underlying values deserve a dedicated table. Clarifications added to caption also in response to referee 2: "Aerosol dry number size distributions ($dN_a/dlogD_a$) reported from HTDMA during the two-hour pre-rain period (colored solid lines; legend indicates Julian date in UTC), lognormal fits to HTDMA (colored dashed lines; text indicates fitted number concentrations in $cm^{-3}$, geometric mean dry diameter in μm and standard deviation), and the final case study distribution derived from the mode-wise linear mean of contributing parameters and its hygroscopicity parameter (κ) derived as the number-weighted mean of contributing HTDMA values (black dashed lines and black text; archived with Supplement 1). In the 20 May case, zero and 8000 $cm^{-3}$ particles in the nucleation mode illustrate BASE and NUCL simulation inputs (dotted black lines)."

Reworded "It is found that two to three modes provide the best fit" to "The Vogelmann et al. (2015) algorithm optimizes a fit of two or three modes" to emphasize that we relied entirely on that algorithm since results appeared consistently satisfactory.

Page 35, Figure 14 – In my opinion, this figure could be removed, and the simulated values added to Table 2.

We have retained it because this figure conveys information that is difficult to fully capture in a table and we removed six other figures.

Page 41, Figure 20 – I am confused by the black BASE trace for number concentration. Doesn't this simulation have a fixed droplet concentration of 250 mg-1, as stated on page 6, line 26?

Correction made to text: "250 $cm^{-3}$."

Page 42, Figure 21 – It is unclear whether only the top left panel is an integrated reflectivity; it seems so given its different scale, but this should be clarified in the caption. A definition of ZHH (as the horizontally-polarized radar reflectivity, right?), along with definitions for the pink, white, and red circles in various subpanels, would help in the interpretation of this figure.

Clarifications added to caption: "Horizontally polarized radar reflectivity ($Z_{HH}$ in dBZ) from KVNX radar (left, dotted red circle): (top) example updraft object at ~12 UTC (solid red) among others identified in units of dBZ km (red-enclosed, see text), (middle) movement of example updraft

from initial location (solid red) towards intersection with the aircraft sampling location (white-enclosed, see text) projected onto 2-km $Z_{HH}$ at ~14 UTC, and (bottom) $Z_{HH}$ curtain obtained from column-wise averages over tracked regions from ~12–15 UTC with Citation ascent legs in time and height (white bars) and averaging time used in Fig. 22 (white lines). From the AERO simulation (right): (top) identification of a typical updraft object projected onto simulated $Z_{HH}$ at ~11 UTC (solid red) among others identified (red enclosed, see text), (middle) its movement from the identified location (solid red) to intersection with the aircraft sampling location (white-enclosed, see text) projected onto simulated 2-km $Z_{HH}$ at ~13 UTC, and (bottom) $Z_{HH}$ curtain obtained from column-wise averages over tracked regions from ~11–14 UTC with mid-point of hour-long averages used in Fig. 22 (white lines)."

Page 43, Figure 22 – "Time 1", "Time 2", etc. have not been defined for the simulations. It would be clearer to label the gray traces 'AERO, Time1' etc. so that the reader knows these are only from that simulation.

Clarification added to caption: "AERO simulation times 1, 2, 3 and 4 indicated in Fig. 21 (light to dark grey lines)."

Technical comments / suggestions Page 3, Line 25 – A term like "droplet activation" or "ice nucleation" or "new particle formation" would more clearly indicate the process(es) meant by "aerosol consumption" here.

Clarification added: "via droplet activation".

Page 3, Line 26 – Remove the second "be".

Removed, thank you.

Page 6, Lines 9 – 10 – Add a 'to': "appears to be variably biased relative to the groundbased measurements".

Added, thank you.

Page 8, Lines 16-22 – Reword through here for clarity, e.g. "Consistent with underestimated Ni, the Dmax at which BASE mass distributions peak is roughly 3-5 times larger than that at which the observed distribution peaks. The Dmax at which the BASE mass distributions peak increases monotonically with increasing mass concentration, whereas the observed mass distributions tend to..

Reworded, thank you.

Page 8, Line 28 – There is an unfinished sentence beginning with "At 6.7 and 7.6 km".

"At 6.7 and 7.6 km. However," corrected to "At 6.7 and 7.6 km, however,"

Page 12, Line 18 – "Updrfts" to "updrafts"

Corrected, thank you.

Page 14, Line 4 – "have" to "has"

Corrected, thank you.

Page 14, Line 6 – "are" to "is"

Corrected, thank you.

Page 22, Figure 1 – It would ease readability if the ARM central facility were marked with a color other than yellow, since the pentagon, bull's eyes, and thumbtacks are all yellow as well.

Agreed. Since this is a stock figure that we did not generate, we did not attempt to adjust it.

Page 24, Figure 3, panel d – Is there a red trace for 0.013 um here? If so, it is not visible.

It is strongly intermittent. Clarification added to caption: "(intermittent at smallest cut)."

Page 33, Figure 12 – It would ease readability if Q2 were expanded to National Mosaic and Multi-Sensor QPE system in this caption, as well as in the text, and again if QPE were expanded here and in the text.

We have now spelled out "National Mosaic and Multi-Sensor Quantitative Precipitation Estimate" in the caption to Figure 12 and in the text.

Pages 36-38, Figures 15-17 – The red and blue traces should be labeled PMS 2DC and HVPS rather than obs1 and obs2.

Both are merged PSDs from the same raw data, adopted here as an estimate of poorly established uncertainty. Clarification added to caption: "Size distributions of ice mass (left) and number (right) in four ranges of ice water content (IWC, ranges in parentheses in g m$^{-3}$) derived from merger of 2DC and HVPS raw data independently by Wang et al. (2015a, 'obs1' in red) and Wu and McFarquhar (2016, 'obs2' in blue). Both are shown as an estimate of poorly established uncertainty."

Page 41, Figure 20 – The y-axis should be '[km]' not '[m]'.

Figure removed.

---

## Author Comment (AC2) · 16 Feb 2017

This study examines and reports aerosol size distribution profiles for six convection case studies observed during the MC3E field campaign, intended for use in model simulation of those cases. The authors demonstrate use of the aerosol size distribution profiles in NU-WRF simulations of the 20 May case study with Morrison twomoment microphysics focusing on examining the stratiform cloud microphysical properties. There are some interesting findings such as ice crystal number concentrations are consistently dominated by a single mode near Dmax of 400 µm, and a mass mode near Dmax of 1000 µm becomes dominant with decreasing elevation to the −10 0C. Therefore, the study is worthy being published. However, this reviewer does have some concerns about the current form as listed below,

We very much appreciate the helpful questions and comments. Point-by-point responses below have greatly improved the manuscript by reducing figures, adding section numbers, and making corrections and clarifications throughout.

(1) I am a little confused about the objectives for the second half of the paper that demonstrates the use of the derived aerosol size distribution. The Introduction does not have a clear statement about the goal of this part. Their results show that simulation using the aerosol size distribution derived does not much affect ice microphysics and stratiform microphysical properties including particle size distribution. These results kind of dispute the importance of aerosol size distribution used in model simulations. Logically, to show the importance of the developed product (i.e., aerosol size distribution), the paper should present results that are significantly changed by aerosol size distribution such as precipitation rate, convection, etc. But the authors did not go to this direction and kind of ignored the point about the importance of the derived aerosol size distribution to MCS simulations. This is ok only if the authors clearly state the reasons for doing what they chose to do and the relevant objectives.

Clarification added to Section 5: "If a warm-temperature ice multiplication mechanism is dominating outflow ice distributions in a manner that cannot be generally reproduced in simulations and is not well understood, it is difficult to confidently assess how or to what degree hygroscopic and ice-nucleating aerosols can be expected to modulate outflow ice properties. For instance, in this study we cannot be confident of the relevance of our sensitivity tests for understanding natural convective outflow owing to inadequate baseline fidelity compared with observations."

(2) Section 3 does not have a clear structure. This part is very important to the entire paper, and the authors need to be clear about (a) the methodology of how the aerosol size distributions are derived, (b) the final products provided to the community, and (c) the discussion about caveats and uncertainties. However, the current writing in this section makes readers difficult to get those. The authors are still talking that the methodology in the last 4 paragraphs of this section.

We now use two levels of subsections in Sections 3 and 4. Some additional text is added for clarification.

(3) The contribution of small CCN to droplet nucleation and ice particle concentration at upper-levels needs some further examination. The conclusion is premature. See comment #20.

Our activation treatment does not omit secondary droplet nucleation above cloud base (see response to comment #20 below). We also now clearly state that the value of our sensitivity tests is limited (see response to comment #1 above).

(4) About Section 5, although I enjoyed reading the discussion, much of the discussion should be moved to the Introduction since they are the very relevant literature studies providing the background for this work.

We consider results unexpected based on past literature, and therefore do not present discussion of results before presenting the results themselves. In the introduction we do mention Ackerman et al. (2015) as a motivating factor.

In addition, some of the things discussed here are not even mentioned in the main text or not much related (for example, the lack of the positive differential radar reflectivity and the importance of the tropical convection in global circulation).

Clarification added to Section 5: "Case studies are generally better for model development if they are relatively typical rather than unusual or rare. ... Analyses of dual-polarimetric radar observations could be further systematically employed to identify the environmental conditions associated with stratiform microphysics regimes ..."

Reference to global circulation now refers back to introduction.

(5) There are many inconsistencies between Figure, Figure captions, and the corresponding text, and also a few figure captions do not clearly describe the figures. There are quite a few sentences what do not make sense or are wrongly stated. Please refer to the specific comments below for the details.

Please see responses below and those to referee 1.

(6) Too many figures: some figures can be combined such as Fig 4 and 5, and some are not key to the main points such as Fig. 9-11, and Fig. 16-17, which could be the options for the supplemental materials since there is already a supplemental file.

We combined Figs. 4 and 5 and removed 6, 10–11, and 19–20. We retained Fig. 9 to show one comparison of derived PSD aloft with observations and 15–17 (main focus).

Detailed comments,

1. P1 Line 14-15, not sure what you want to say here, especially about the specific meaning of "the microphysics pathways associated with deep tropical convection outflow".

Reworded for clarification: "Based on several lines of evidence, we speculate that updraft microphysical pathways determining outflow properties in the 20 May case are similar to a tropical regime, likely associated with warm-temperature ice multiplication that is not well understood or well represented in models."

2. P2 Line 2, aerosol should be plural here.

Changed.

3. P2 Line 10-14, this is a very long sentence. Suggest to break into two sentences to make it easier to read.

Done.

4. P2, last paragraph, the last a few sentences of this paragraph need to be revised to clearly state the objectives of this study. If the objective is to achieve more accurate simulations, then is the goal achieved?

With respect to the last four sentences in this paragraph, we achieve the goals stated in the first to third, which respectively begin "Here we" and "We also". The last sentence begins "Enabling accurate simulation" because we intend the derived aerosol PSDs for that purpose. Since the latter is better discussed in Section 5, we removed the last sentence.

5. P3 Line29, aerosol should be plural here.

Changed.

6. P4, Line 5-11: the description here about Figure 3 suggests Na is from DMA or CPC and kappa is from HTDMA. However, the Figure 3 caption said only HTDMA, and no DMA data is shown. Please clarify the inconsistency. In addition, description about instrumental uncertainty for each instrument would be helpful here.

HTDMA now used consistently throughout. Clarification added to Section 3.2: "Based on the discrepancy between ground-based CPC and HTMDA measurements, we estimate that overall uncertainty in derived total aerosol number concentrations is roughly a factor of two throughout this work."

7. P4, Line 15-16, something is missing in the later half of the sentence. Otherwise, it does not make sense.

Latter half simplified to "nucleation mode aerosols were commonly present in large concentrations but were also commonly absent."

8. P4, Line 15-19, the description here would be clearer if the ratios of CCN to CPC aerosol concentrations are shown.

Agreed, but since we only show CCN data for completeness (not used in our fits) and we list values in Fig. 3a, we prefer to briefly state the range of ratios rather than adding another figure panel.

9. P5, Line 8 and Lin 17: what are non-case-study dates and case study dates?

Figure and sentence removed (Section 2 describes case study selection).

10. P5, I do not understand what is said in the sentence "UHSAS/CPC again sometimes decrease, not because UHSAS decreases but because CPC increases, consistent with evidence that the surface is also a source of fine particles". CPC increases suggested more small particles, which could be from particle nucleation at the elevated altitudes. This is observed quite often. So, I do not understand why we can infer that surface is the source.

Sentence clarified: "However, the local minimum in the ratio of UHSAS to CPC seen at the surface is consistent with a surface source also for fine partices (e.g., Wang et al., 2006, their Fig. 7), which could be both spatiotemporally variable and regional in nature (e.g., Crippa et al., 2013)."

11. Figure 7, there are no red and blue lines.

Figure corrected and caption revised also in response to referee 1: "The median of airborne CPC and UHSAS aerosol number concentrations within 1-km-deep layers for each MC3E flight, and the ratio of those median values for the seven flights with both instruments (black lines). The median of profile values at each elevation (red lines) are archived as Supplement 2."

12. Figure 8, why are there two colored solid lines for the measurement from HTDMA? It is really confusing with so many numbers on each panel and the description is not clear for some numbers such as the numbers at the right bottom part of each panel. Strongly suggest to use a table to show the parameters for the three modes. Also, need to explain the purpose of showing the 0 and 8000 cm-3 in the nucleation mode for May 20 case.

The black values are archived with Supplement 1 and we disagree that the underlying values deserve a dedicated table. Clarifications added to caption also in response to referee 2: "Aerosol dry number size distributions ($dN_a/dlogD_a$) reported from HTDMA during the two-hour pre-rain period (colored solid lines; legend indicates Julian date in UTC), lognormal fits to HTDMA (colored dashed lines; text indicates fitted number concentrations in cm$^{-3}$, geometric mean dry diameter in μm and standard deviation), and the final case study distribution derived from the mode-wise linear mean of contributing parameters and its hygroscopicity parameter (κ) derived as the number-weighted mean of contributing HTDMA values (black dashed lines and black text; archived with Supplement 1). In the 20 May case, zero and 8000 cm$^{-3}$ particles in the nucleation mode illustrate BASE and NUCL simulation inputs (dotted black lines)."

13. Fig. 8, there are such large differences in the measurements of HTDMA for 4/25 and 5/24 in the smallest mode (although it is not clear each colored solid line represent), then any fit should have very large uncertainty. Is it meaningful for such a fit?

Clarification added to Section 4.1: "Since nucleation-mode aerosol (in the smallest fitted mode) are present very non-uniformly in time and space during some MC3E case studies (cf. Fig. 6), we finally test whether that is likely to be important."

14. Fig. 9, what is N? What is total aerosol number size distribution?

Clarification added to figure and caption: "Derived modes and aerosol number size distribution over 1-km-deep layers (black dotted and dashed lines, respectively) compared with bin-wise mean and median out-of-cloud UHSAS size distributions (red and blue lines, respectively) for the 25 April case study, with sample size (cf. Fig. 4) and total aerosol number concentration ($N_a$) in cm$^{-3}$."

15. P6 Line 27-32, the text here is confusing: first, need to be specific about aerosol configurations in AERO. It is not enough to just say "initialized with the aerosol profile described above" since it is not clear "above". To me, Fig. 8 is above but there are many different aerosol parameters listed on the panel for 5/20.

Clarification added: "Aerosol are initialized within all domains to the 20 May aerosol input profile derived as described in Section 3.4 (see Supplement 1), and are fixed to it at the outermost domain boundaries."

Second, since AERO has prognostic droplet number concentrations, I do not understand why need to fix droplet number concentrations at the boundary? Shouldn't fixing aerosol be enough?

Clarification added per response to comment #16.

Third, I do not understand "Unknown aerosol source terms are neglected", thus I am confused with the later part if the sentence "how all else being equal, this increases the difference between BASE and AERO results".

By unknown we meant that aerosol source terms cannot be readily observed and specified. Simplification and clarification made also in response to referee 1: "Aerosol source terms beyond advection across outer domain boundaries are neglected (e.g., primary emission and gas-to-particle conversion)."

Lastly, it is not clear what cloud microphysics scheme is used for other simulations besides BASE.

Clarification added: "We compare observed hydrometeor size distribution properties with those simulated using Morrison et al. (2009) two-moment microphysics with hail." Additional detail is then added on the ice nucleation parameterizations used throughout (mostly off in HOMF).

16. P6 Line 33, BASE should have no aerosol since droplet number is not prognostic as shown in Table 1.

Clarification added: "In the baseline simulation (BASE), we use a fixed droplet number concentration of 250 cm$^{-3}$. In the AERO simulation, droplet number concentration is treated prognostically as follows."

17. P7 Line 1-2, why 8000 cm-3? This sounds a very large aerosol number concentration.

Reference added and clarification also in response to referee 1: "Based on the April and 1 May nucleation-mode fits listed in Fig. 6, this represents the most commonly fit mode diameter and rounded mode standard deviation, and a modest number concentration (maximum on 1 May) that is lower than typically observed in the 10–30-nm diameter range during intense new particle formation events (e.g., Crippa and Pryor, 2013)."

18. P7, the third paragraph and Fig. 12: Q2 and Q2corr cover the entire domain, why not compare the precipitation over the entire domain? Suggest to add such a plot to Fig. 12 (after all, it would be a more robust comparison compared with that over a small domain of 100x100 km2).

We illustrate observed and simulated precipitation rates over the entire domain in Figs. 9 and 16 for context, but the objective of Fig. 8 is to show the observed and simulated time series specifically within the aircraft sampling domain that is also used for the comparisons of stratiform ice and rain properties. Clarification added to caption: "averaged over the region sampled by aircraft after 13 UTC indicated by a red rectangle in Fig. 9."

19. Figure 14, There is only one observation dataset shown in the figure, why are there two sources (Wang etal. 2015a and Wu and McFarquhar 2016)? The related discussion about the two measurements is on P8 Line 9 but the figure does not show both.

The box and whisker plots contain both observational data sets. Caption simplified" "from aircraft observations (left, see text) and from the BASE simulation (right)". Clarification added Section 4.2.1: "Fig. 10 shows ice water content (IWC) and ice number concentration (Ni) from both independently derived observational data sets."

20. P9 Line 12-14, If Morrison scheme is used, do you consider second droplet nucleation or only cloud-base nucleation is considered? I would expect secondary nucleation at higher altitudes could make significant differences if small CCN is present. Therefore, I would suggest to do another test with the secondary nucleation considered if it is not considered in the NUCL.

Clarification added to Section 4.1: "Aerosol activation follows the treatment of Abdul-Razzak and Ghan (2000), in which the supersaturation is taken as the minimum value over the time step following Morrison and Grabowski (2008, their Eqn. A10), as in Vogelmann et al. (2015, see their Sect. 5.1)." This approach does not limit droplet activation to cloud base.

21. P9 Line 18-20, I think the point is mainly supported by much smaller ice particle number concentration simulated by the model.

We consider uncertainty in observed particle number concentration far greater, as emphasized in the last sentence of the following paragraph.

22. Figure 21, please define Zm and ZHH. Also, I do not understand why each panel is plotted for a different time? And the figure order does not reflect a time evolution, and the color legend is different for the same type of figures between observation and model simulation such as Panels 2 and 3. What does the red color denote in the first four panels?

Clarifications added to caption also in response to referee 1: "Horizontally polarized radar reflectivity ($Z_{HH}$ in dBZ) from KVNX radar (left, dotted red circle): (top) example updraft object at ~12 UTC (solid red) among others identified in units of dBZ km (red-enclosed, see text), (middle) movement of example updraft from initial location (solid red) towards intersection with the aircraft sampling location (white-enclosed, see text) projected onto 2-km $Z_{HH}$ at ~14 UTC, and (bottom) $Z_{HH}$ curtain obtained from column-wise averages over tracked regions from ~12–15 UTC with Citation ascent legs in time and height (white bars) and averaging time used in Fig. 22 (white lines). From the AERO simulation (right): (top) identification of a typical updraft object projected onto simulated $Z_{HH}$ at ~11 UTC (solid red) among others identified (red enclosed, see text), (middle) its movement from the identified location (solid red) to intersection with the aircraft sampling location (white-enclosed, see text) projected onto simulated 2-km $Z_{HH}$ at ~13 UTC, and (bottom) $Z_{HH}$ curtain obtained from column-wise averages over tracked regions from ~11–14 UTC with mid-point of hour-long averages used in Fig. 22 (white lines)."

23. P10 Line 5-6, why suddenly talking about BASE since only AERO is compare with observations in both Figures 21 and 22.

Corrected, thank you.

24. P10 Line 30-31, suggest to reword the sentence. It is not easy to understand currently.

Agreed, reworded: "We note that breakup equilibrium is thought to require rain rates on the order of 50 mm h$^{-1}$, substantially greater than typical of stratiform regions (e.g., less than 15 mm h$^{-1}$ in Fig. 8), but its existence, size distribution characteristics, and prevalence in nature have been elusive (e.g., McFarquhar, 2010; D'Adderio et al., 2015)."

25. P11 Line 15-16, "we find that predicted and observed stratiform ice size distributions are similarly coherent within the stratiform region": I am not sure what this sentence really means since simulated and observed size distributions are totally different as shown in Figs. 14-17.

Clarification added also in response to referee 1: Reworded to "both predicted and observed stratiform ice size distributions exhibit relatively well-defined properties that do not vary rapidly in time."

26. The third paragraph in Section 5: this paragraph summarizes observed results. It is natural to comparatively describe how model does here, and this information is missing from the summary currently.

Simulated number concentration and peak of ice mass size distribution are summarized in the last sentence of the second paragraph. Added there re sensitivity tests: "Results are insensitive to prognosing droplet number concentration using an observation-based profiles with or without nucleation-mode aerosol (in place of fixed droplet number concentration). Additionally turning off all ice nucleation and multiplication parameterizations except homogeneous cloud droplet and raindrop freezing leads to less and larger ice."

Added to the end of the third paragraph : "In simulations, unlike in observations, the $D_{max}$ where the mass size distribution peak increases substantially with mass concentration at each elevation (where there is more ice mass, it is also systematically larger) and the number concentration decreases rapidly with elevation. Beneath the aircraft-sampled region, simulated mass-weighted mean diameter of rain is roughly 0.7 mm larger than retrieved, consistent with overlying ice size bias; collocated reflectivity within the range observed is consistent with a corresponding low bias in precipitation rate (Fig. 8)."

---

## Referee Report (RR1)

Review 2 of *Use of an observation-based aerosol profile in simulations of a mid-latitude squall line during MC3E: Similarity of stratiform ice microphysics to tropical conditions*

Thank you for addressing my concerns. The manuscript is much improved, particularly organization and readability. It is ready for publication in my opinion. I make a few, minor suggestions below.

**Specific comments / suggestions**

Page 4, Lines 20-24 – The first sentence could be rewritten for clarity, e.g., "The large spread between CCN, HTDMA, and ground-based CPC measurements reflect the large variability in nucleation mode aerosol concentrations." To me, it also makes more sense to place this paragraph along with the first to separate discussions of number concentration and hygroscopicity.

Page 6, Line 1 – I would include the "user-determined size" employed in this case for nucleation mode truncation.

Page 9, Line 29; Page 10, Line 5 – -10°C / -16°C are the mean temperatures at these altitudes? Please include $\bar{T}$ in the parentheses.

Page 11, Lines 5-6 – The Bigg 1953, Meyers et al. 1992, and Cooper 1986 schemes which are used for ice nucleation are known to have limitations since they do not account for spatiotemporal variation in INP, e.g. Prenni et al. BAMS 2007 or DeMott et al. PNAS 2011. This point could be made here in the note that "critical aspect of ice nucleation" may be missing.

Page 30, Figure 5 caption – Reword second sentence for clarity: "The campaign-wide median profiles at each elevation (red lines) are archived as Supplement 2."

Page 31, Figure 6 – I still struggle to interpret all of the information in this figure. Given that the distributions in each mode are ultimately averaged in time, could these average number concentration, geometric mean dry diameter, and standard deviation values be presented? And the colored values from various Julian dates omitted or moved to Supplemental Information?

Page 36, Figure 11 right hand panels – Would it make comparison of the BASE simulation and observed number size distributions easier if the y scale were logarithmic?

Page 46, Table 2 caption – It would be helpful to indicate that the range in these values comes from the fact that they are calculated from "five level legs between 13.9 and 14.9 UTC" (otherwise, it seems as if there should be just two discrete values from the two methods).

---

## Author Response (AR2)

Thank you for addressing my concerns. The manuscript is much improved, particularly organization and readability. It is ready for publication in my opinion. I make a few, minor suggestions below.

We appreciate the helpful suggestions. Point-by-point responses follow.

Specific comments / suggestions

Page 4, Lines 20-24 – The first sentence could be rewritten for clarity, e.g., "The large spread between CCN, HTDMA, and ground-based CPC measurements reflect the large variability in nucleation mode aerosol concentrations."

Change made, with slight revision to "The large variability of spread… reflects…"

To me, it also makes more sense to place this paragraph along with the first to separate discussions of number concentration and hygroscopicity.

Paragraph moved.

Page 6, Line 1 – I would include the "user-determined size" employed in this case for nucleation mode truncation.

Clause removed (no truncation is used in this study).

Page 9, Line 29; Page 10, Line 5 – -10°C / -16°C are the mean temperatures at these altitudes? Please include T̄ in the parentheses.

Elevation-wise temperature now expressed as median and range in Table 2, and median to two significant figures is retained throughout text for readability.

Page 11, Lines 5-6 – The Bigg 1953, Meyers et al. 1992, and Cooper 1986 schemes which are used for ice nucleation are known to have limitations since they do not account for spatiotemporal variation in INP, e.g. Prenni et al. BAMS 2007 or DeMott et al. PNAS 2011. This point could be made here in the note that "critical aspect of ice nucleation" may be missing.

Clarification added above to Section 4.1: "All of these heterogeneous ice nucleation parameterizations neglect spatial variability and consumption of ice nucleating particles."

Page 30, Figure 5 caption – Reword second sentence for clarity: "The campaign-wide median profiles at each elevation (red lines) are archived as Supplement 2."

Change made, with omission of "at each elevation" for further simplification.

Page 31, Figure 6 – I still struggle to interpret all of the information in this figure. Given that the distributions in each mode are ultimately averaged in time, could these average number concentration, geometric mean dry diameter, and standard deviation values be presented? And the colored values from various Julian dates omitted or moved to Supplemental Information?

Contributing fits and associated text are now omitted from Fig. 6. Input size distribution data (colored solid lines) are shown alongside the fitted values (black dashed lines, included in Supplement 1) to illustrate range and quantity of underlying values.

Page 36, Figure 11 right hand panels – Would it make comparison of the BASE simulation and observed number size distributions easier if the y scale were logarithmic?

We prefer dX/dlogD versus logD on a linear vertical scale because that yields visually integrable plots that are most intuitive to understand.

Page 46, Table 2 caption – It would be helpful to indicate that the range in these values comes from the fact that they are calculated from "five level legs between 13.9 and 14.9 UTC" (otherwise, it seems as if there should be just two discrete values from the two methods).

Added "by flight leg elevation" without times since a few values come from later profiles (see Section 4.2.1).

**Response to second interactive comment on "Use of an observation-based aerosol profile in simulations of a mid-latitude squall line during MC3E: Similarity of stratiform ice microphysics to tropical conditions" by Ann M. Fridlind et al. by Anonymous Referee #2**

The authors addressed the majority of my comments in the previous round.

We appreciate this second review.

However, there are a few comments that need further work as detailed below,

(1) A table for the data listed on each panel in Figure 6: I noticed both reviewers of the paper suggested this but the authors did not follow the suggestion. Currently, with so many values on the panels (some data are even overlapped with the lines of the plots), it is simply too much and difficult to focus on anything. If the values are provided in the supplemental material, then refer to the supplemental material and remove from the panels. If not, then have a table listing them would be clearer (also make the plots neater).

Contributing fits and associated text are now omitted from Fig. 6. Only input size distribution data (colored solid lines) are shown alongside the fitted values (black dashed lines, included in Supplement 1) to illustrate range and quantity of underlying values.

 (2) About my major comment #1, it seems that the authors' response is not for this comment. By reading the revised manuscript, I still have the similar concerns. The title and introduction are not well aligned with the objectives and the content of this study and need further clarification. Basically the paper consisted of two parts: retrieved aerosol SD for MC3E cases and examine the stratiform ice microphysics from a single case study with and without the retrieve aerosol SD. The current introduction is mainly for the second part of the study and essentially no context/background for the first part (I only see one sentence related to it at line 27-29 on P2). For readers who are not in the aerosol-cloud interaction fields, they might not know the connections between aerosols and cloud properties. Therefore, context/background (such as the general relationships of aerosols with clouds and the significance of the aerosol impact could be) is needed for part 1. In part 2, since the authors show that the biases of stratiform ice microphysics is not depending on aerosol used, discussion about that the biases might not caused by aerosol impact (at least for the synoptic system simulated in this study) would be useful. The current title is worded as "Similarity of stratiform ice microphysics to tropical conditions", which makes readers anticipate the comparison results between the simulated case and a tropical case. However, no results are shown for this point, and this is only a discussion point in Section 5. Therefore, the title needs to be changed to better aligned with the results of the paper.

The introduction refers interested readers to several recent reviews of aerosol-convection interactions and motivates derivation of aerosol input profiles (explaining why we embarked on this study). We revised the title to "Derivation of hygroscopic aerosol input profiles for MC3E convection studies and use in simulations of the 20 May squall line case," which no longer highlights the most interesting results of the paper (which were unexpected and cannot be properly discussed until after the results are presented) but is more consistent with its structure.

(3) About secondary nucleation in Morrison scheme (my comment #20), the scheme does have an option for separation of cloud base nucleation from secondary nucleation. The default option is for cloud base nucleation only (i.e., the supersaturation parameterized in ARG scheme nucleation is only for grid points without pre-existing cloud). To consider secondary nucleation, supersaturation calculation needs to be modified to consider the condensational growth of the existing droplets. Yang et al. (2015) has the details to add the secondary ice nucleation into Morrison scheme (Yang et al. 2015, Aerosol transport and wet scavenging in deep convective clouds: A case study and model evaluation using a multiple passive tracer analysis approach, J. Geophys. Res. Atmos., 120, doi:10.1002/2015JD023647).

We implemented in NU-WRF a scheme from DHARMA that includes mode-wise aerosol consumption and transport and explicit calculation of condensational growth (see published uses and description in Section 4.1). Additional clarification added: "
[revised manuscript text omitted]
 | $-$11 | $-10.2$–$-11.7$ | 0.52–0.56 | 0.89–1.0 | 45–46 | 72–80 | 948–993 | 1850–2200 |